# Quantitative evidence for dimorphism suggests sexual selection in the maxillary caniniform process of *Placerias hesternus*

James L. Pinto[1]*, Charles R. Marshall[1], Sterling J. Nesbitt[2], Daniel Varajão de Latorre[1]*

**1** Department of Integrative Biology and Museum of Paleontology, University of California, Berkeley, California, United States of America, **2** Department of Geosciences, Virginia Tech, Blacksburg, Virginia, United States of America

\* jameslpinto@berkeley.edu (JLP); daniel.latorre@berkeley.edu (DVL)

**Data Availability Statement:** All relevant data are within the manuscript and its Supporting information files.

## Abstract

*Placerias hesternus*, a Late Triassic dicynodont, is one of the last megafaunal synapsids of the Mesozoic. The species has a tusk-like projection on its maxillary bone, known as the caniniform process. This process has been hypothesized to be sexually dimorphic since the 1950s, however this claim has not been thoroughly investigated quantitatively. Here, we examined maxillae, premaxillae, quadrates, and fibulae from a single population from the *Placerias* Quarry in the Blue Mesa Member of the Chinle Formation, near St. Johns, Arizona, USA to determine if the caniniform process is dimorphic. We made a total of 25 measurements from the four bones and used a maximum likelihood framework to compare the fit of unimodal versus bimodal distributions for each set of measurements. Our results from complete maxillae reveal that the caniniform process has two distinct morphs, with a shorter and longer form. This interpretation is substantiated both by strong statistical support for bimodal distribution of caniniform lengths, and by clustering analysis that clearly distinguishes two morphs for the maxillae. Clustering analysis also shows support for potential dimorphism in the shape of the quadrate. However, no measurements from elements other than the maxilla have a strong likelihood of bimodal distribution. These results support the long-standing hypothesis that the caniniform in *Placerias* was dimorphic. Alternative explanations to sexual dimorphism that could account for the dimorphism among these fossils include the presence of juveniles in the sample or time-averaged sampling of a chronospecies, but both have been previously rejected for the *Placerias* Quarry population. The lack of strong dimorphism in non-maxilla elements and increased variation in caniniform length of the large-caniniform morph suggest that the caniniform is a secondary sexual trait, possibly used in intraspecific competition.

## Introduction

### Sexual dimorphism in dicynodonts

Sexual dimorphism is notoriously difficult to establish in the fossil record [but see 1, 2], especially in vertebrates, unless there are massive sample sizes [3]. The detection of sexual size

**Funding:** Amended Funding Statement: CRM was partially supported by the Philip Sandford Boone Chair in Paleontology at the University of California, Berkeley. There was no additional external funding received for this study.

**Competing interests:** The authors have declared that no competing interests exist.

dimorphism (SSD) has been particularly challenging because the upper and lower extremes of each sex's size distribution often overlap, making it hard to distinguish the two even for extant taxa known to be sexually dimorphic, particularly if *a priori* knowledge of sex is removed [4]. In contrast, discontinuous secondary sexual traits, such as sexually dimorphic weaponry or ornamentation used for intraspecific combat or competition through visual display, are typically clearly dimorphic, either being significantly enlarged in, or entirely unique to, one sex [5]. Many living synapsids have features that fall into this second category, like the presence or size of a tusk or horn [6], and as a result the sex of an individual can be distinguished even from isolated skeletal elements. These structures sometimes continue to grow past sexual maturity in the competitive sex [7], usually males in living synapsids [8], which can lead to a positively allometric relationship between body size and secondary sexual structure size.

Tusks have evolved to become sexually dimorphic multiple times in synapsid groups, including in elephants, walruses, and muntjacs. The oldest known occurrence of this adaptation is in dicynodonts, a clade of non-mammalian therapsid synapsids. Genera like the Late Permian *Diictodon* have been demonstrated to have a sexually dimorphic presence or absence of tusks [9]. Many other dicynodonts have been argued to have sexually dimorphic weaponry or ornamentation, ranging from tusks to nasal bosses to varying robustness of facial bone structures such as the maxillary caniniform process. Sexual dimorphism in dicynodonts was first suggested in the 19th century in *Lystrosaurus murrayi* [10], and variation in cranial elements based on sex was proposed soon after [11]. Arguments for dimorphism have sometimes been quantitative, for example in *Aulacephalodon* [12] and *Lystrosaurus* [13], but have often been based on just qualitative differences among a handful of specimens, and thus need more data or analysis to be statistically supported. Examples include *Dinodontosaurus* [14], *Tetragonias* [15], *Stahleckeria*, *Ischigualastia* [16], *Pelanomodon* [17], and *Wadiasaurus* [18]. *Wadiasaurus* has also been speculated to have formed nursery herds of females and juveniles, similarly to modern elephants, based on the presence of an assemblage site with only tuskless and juvenile individuals, though remains from this taxon are mostly fragmentary, making these claims difficult to assess.

## Dicynodonts in the Late Triassic

One of the earliest radiations of herbivorous megafauna following the end-Permian mass extinction was the kannemeyeriiform dicynodonts, a clade that ranged from the Induan in the earliest Triassic [19] to the Late Norian or Early Rhaetian [20, 21] near the end of the Triassic. The last and largest members of this group all fall within the Stahleckeriidae [22], and are the latest Mesozoic occurrences of synapsid megafauna, with later synapsids (in the form of mammals) not reaching similar sizes until more than 140 million years later [23]. Stahleckeriidae has been viewed as a "relict taxon" of the Kannemeyeriformes, which were prolific throughout the Triassic [22], though the Stahleckeriidae was still taxonomically diverse and distributed across at least four continents in the Late Triassic [20, 24–26]. However, most species are only known from a few individuals, most assemblages where they occur only bear one species [22], and they are less abundant than other contemporaneous groups such as the archosaurs. The only valid kannemeyeriiforms described from North America are the placeriine stahleckeriid *Placerias hesternus* [24], the only species in the genus *Placerias*, *Eubrachiosaurus browni* (a stahleckeriine stahleckeriid) [22, 27], and *Argodicynodon boreni* (a placeriine stahleckeriid) [28], all of which are extremely rare in their respective known areas. The first of these taxa to be described, and by far the most well-known, is *Placerias hesternus*.

*Placerias hesternus* (hereafter referred to simply as *Placerias*) is one of the largest known species of dicynodonts, and among the most massive herbivores known from the Chinle

Formation, reaching three meters in length and possibly weighing over one tonne [29]. *Placerias* has a farily broad geographic distribution, having been found in the Chinle Formation in Arizona [30] and New Mexico [31], the Pekin Formation in North Carolina [32], and possibly the Dockum Group in Texas [33], but it is mostly known from fragmentary remains. The one exception is the *Placerias* Quarry, a small death assemblage site in the Blue Mesa Member of the Chinle Formation, near the boundaries of Petrified Forest National Park, southwest of the town of St. Johns, Arizona, USA, which has produced over 1700 elements from at least 41 individuals of *Placerias* [34]. As a result, *Placerias* provides a unique opportunity to observe individual variation in a single population of large Late Triassic dicynodonts, not offered by the other, rarer stahleckeriids.

The initial description of *Placerias* was over a century ago [24], but it was only described from a humerus until a comprehensive monograph of its anatomy was published based on material from *Placerias* Quarry [29]. The site, though extremely dense in bone material, is almost entirely composed of disarticulated bones, including separated skull bones. Camp and Welles' monograph included a composite reconstruction of the skull of *Placerias*, which was physically constructed from bones of at least nine different individuals [29], with gaps filled in using plaster. The composite skull was created with the largest left and right maxillae from the collection (UCMP 24935 and PEFO 2369, respectively), and was meant to represent an individual among the largest in the population. The reconstructed skull was modified in a later publication by Cox [14], with revisions including replacing the maxillae with slightly smaller ones (UCMP 25317 and UCMP 25318), and changing their angle to improve articulation with the premaxilla (Fig 1). An articulated partial skull of *Placerias* (MNA.V.8464) was later described from a different locality and supports Cox's interpretation of the position of the maxilla, but it is poorly preserved and crushed, making anatomical analysis difficult [22, 31].

The composite skull and measurements from multiple specimens for each cranial element [29] show that *Placerias* had an extremely robust skull, with very large squamosals and jugals connected to rugose maxillae and premaxillae that were likely keratinized, suggesting a herbivorous diet of tough plant remains such as roots and tubers [35]. Like most dicynodonts, *Placerias* has only two teeth: a set of true tusks. While tusks evolved independently in multiple

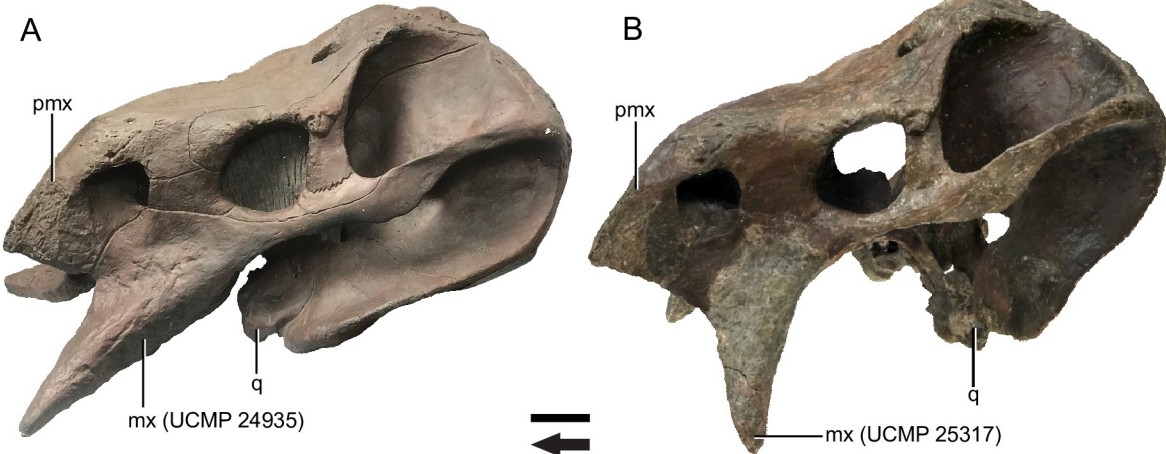

**Fig 1. Composite skull of *Placerias* (UCMP 137369).** (A) photograph in left lateral view of a plaster cast based on the original composite reconstruction from Camp and Welles [29]; (B) the current version of the composite as revised in Cox [14]. Primary differences include shifting the maxilla and quadrate posteriorly in B, and the replacement of the maxillae in A with smaller elements. *Abbreviations*: *mx*, maxilla; *pmx*, premaxilla; *q*, quadrate. Arrow denotes anterior direction. Scale bar equals 10 cm.

dicynodont groups [36], those found on *Placerias* and other kannemeyeriiforms lack enamel and have a cone-in-cone growth arrangement [29, 37], much like the tusks of living mammalian synapsids such as walruses and elephants. *Placerias*, like all dicynodonts [38], has a projection of rugose bone on the maxilla lateral to the tusk, known as the "caniniform process". The size and shape of the caniniform and the nearby tusk vary widely across the clade, with some species lacking tusks entirely. In *Placerias* the caniniform process forms a blade-like edge, particularly on the anterior surface ventral to the premaxillary suture. The posterior surface of the maxilla is wider mediolaterally and lacks this edge. The edge meets with the ventral edge of the premaxilla, creating a single continuous "beak", similar to what is seen in most other stahleckeriids [16]. Unique to *Placerias*, however, is the extreme length of the caniniform, the tip of which lies ventral to the closed mandible in some individuals.

In their description of the *Placerias* Quarry material, Camp and Welles [29] identified two potential morphs among isolated maxillae, one with a much larger caniniform than the other, occurring in roughly equal numbers. From this observation, they hypothesized that *Placerias* was sexually dimorphic and suggested that the large-caniniform morph consisted of males while the small-caniniform morph represented females. However, this suggestion was based on a qualitative assessment only, and they did not provide a justification for the attribution of sexes to the morphs. Here, we statistically test for the presence of two morphs in the maxillae in the population from *Placerias* Quarry. Additionally, we ask whether there is any quantitative indication for the presence of two morphs in three other well-represented bones (the fibula, quadrate, and premaxilla) in the fossil assemblage. Tusks in dicynodonts have been speculated to be used for feeding, digging, and display [36]. In *Placerias*, we surveyed the presence and morphology of tusks in maxillae, and the morphology of dissociated tusks, to test the hypothesis that the caniniform replaced the tusks for these functions, and discuss evidence that the tusk is not under strong selection or essential for individual survival.

## Materials and methods

### Specimen assessment

To quantitatively test Camp and Welles' [29] hypothesis of dimorphism in the maxilla of *Placerias*, we analyzed a total of 36 isolated maxillae from the *Placerias* Quarry population housed in the UCMP and PEFO. Furthermore, to test if other skeletal elements of *Placerias* show evidence of dimorphism, we measured 29 premaxillae, 44 quadrates, and 17 fibulae (Full lists of specimen numbers are available in S1 Table). No permits were required for the described study, which complied with all relevant regulations. We selected these elements because they have the largest sample sizes of bones complete enough to have identifiable homologous landmarks for making linear measurements. While both the premaxilla and the quadrate are cranial bones, the former articulates directly with the maxilla, and the latter does not. Hence, if there is dimorphism in the maxilla and it affects bones in its immediate vicinity, this would more likely be observed for the premaxilla than for the quadrates. Fibulae were included as the largest sample of a postcranial long bone, particularly because long-bone cross-sectional diameters are useful for estimating body size [39], and could reveal size dimorphism in *Placerias*.

Our sample of 36 maxillae (20 left maxillae, 16 right maxillae) has some overlap with the specimens analyzed by Camp and Welles [29]. However, the level of overlap is difficult to ascertain, because while they mention a total count of 39 maxillae (21 lefts and 18 rights) only 17 of those were referred to by a specimen number. Most of these were included in a table of 15 specimens, where they provided measurements of four traits. Two of those 15 specimens had specimen numbers that do not match any in the UCMP collection or database and thus are likely typos in their table. Under this assumption, the specimens they identify as UCMP

27319 and 28398 may well be UCMP 25319 and 28389, respectively. In addition to our sample of 36 maxillae, we found five unlabeled fragmentary maxillae, two of which had been sectioned. An additional 12 maxillae are referred to in the UCMP collections database, five of which have been reported as missing from the collection for decades, and we were not able to find any of these specimens in the collection. One maxilla (UCMP 27370) was figured by Camp and Welles [29], but is not in the UCMP database and was not located in the collection. Other than UCMP 27370 and the two potentially misreferred specimens, all maxillae mentioned by specimen number by Camp and Welles [29] are used in this study. Another maxilla (PEFO 2639) was relocated to Petrified Forest National Park for display purposes, but has a field number consistent with the *Placerias* Quarry specimens, and was evidently part of the original composite skull before being removed by Cox [14] in his modifications, based on its similarity to the maxilla in the cast of the original composite and the presence of a cut iron rod in the specimen that had been used to attach it to the rest of the skull. All of these labeling discrepancies complicate tracking the exact number of found maxillae, but in total, 49 maxillae collected by the UCMP from *Placerias* Quarry have been referred to in the past or otherwise theoretically exist, from at least 21 individuals, which falls well within the 41 minimum number of individuals reported from the site [34], based on left postorbitals, which though common are often fragmentary. Our work is based on a sample of at minimum 20 individuals, based on left maxillae. A list of specimen numbers of all specimens used is available in the S1 Table.

Within this sample, the quality of preservation is very uneven. Particularly, material from the site has suffered from diagenetic fracturing and from excavation or preparation related breakage (e.g, "marks of discovery" and other toolmarks) [34]. As a result, some of the maxillae are not fully intact, but they were included in the analysis if they were complete with respect to the specific measurement being taken. Extremely fragmentary specimens were considered only for the evaluation of presence/absence of tooth sockets. Determining which left and right elements come from the same individual is extremely difficult due to the disarticulated nature of the *Placerias* Quarry material. Camp and Welles [29] stated that some maxillae can be putatively paired, but there is no evidence for the proposed pairs beyond their roughly similar size and shape. Though some association between elements like osteoderms has been found in aetosaurs from *Placerias* Quarry [40], based on plotting their locations in positional grid squares, only one pair of maxillae (both numbered UCMP 27553) were found in the same grid square. These are of a similar size, and thus likely associated, but were not actually found connected, and so cannot be confirmed as a pair.

Similarly to the maxillae, many of the 29 premaxillae (24 fused pairs, four right fragments, one left fragment) in the UCMP collection were fragmentary, and only 9 were usable for complete length measurements. The 44 quadrates (23 lefts, 21 rights) also had varying levels of intactness, with 26 being complete enough for all length measurements, and 40 having an intact medial mandibular condyle, the proportions of which can be used as a rough proxy for quadrate size. Eight of the 17 fibulae were complete enough for all length measurements, and a further six had the proximal condyles present, the dorsoventral lengths of which can be used as a proxy for length of the overall fibulae.

## Measurements

Maxillae were measured along the dorsoventral length from the dorsal tip of the jugal suture to the ventral tip of the caniniform process, or jugal-caniniform process dorsoventral (jcp DV) length, mediolateral width from the lateral tip of the jugal suture to the furthest medial point just beneath the pterygoid pit, or jugal-pterygoid pit mediolateral (jpp ML) width, and

proximal anteroposterior length from the anterior tip of the premaxillary suture to the posterior tip of the jugal suture, or jugal-premaxillary anteroposterior (jpm AP) length. Tooth diameter (td) was also measured if the tooth was present, along with the depth of the large cavity proximal to the alveolus, or alveolar cavity depth (ad), as well as distance from the tooth socket (if present) to the caniniform tip (ttc), giving a total of 6 length measurements. The cavity proximal to the alveolus referred to here is a smooth conical depression on the posterior portion of the dorsal face of the maxilla, in the position of the dental chamber in other kannemeyeriiforms, and likely served a similar function. In some other kannemeyeriiforms (e.g. *Kannemeyeria*), the alveolus and proximal end of the tooth is commonly exposed in the dental chamber, however in *Placerias* this has been reported to be much less common among individuals [29]. To assess this claim, we also recorded three nominal measurements relating to the tooth that were observed to vary among specimens: presence or absence of the tooth, presence or absence of tooth socket, and eruption of the tooth from the proximal end of alveolus (Fig 2). Seven disassociated tusks were observed for presence of features including longitudinal grooves or variation in diameter across the root, to assess potential variation within the population.

For the premaxillae, the following eight measurements were recorded: anterior beak tip to posterior nasal process tip (at to np), anterior tip to anterior end of maxillary suture tip (at to mxs), mediolateral length of palatal groove (pg ML), anterior end of maxillary suture tip to nasal process tip (mxs to np), anteroposterior length (AP), mediolateral length between

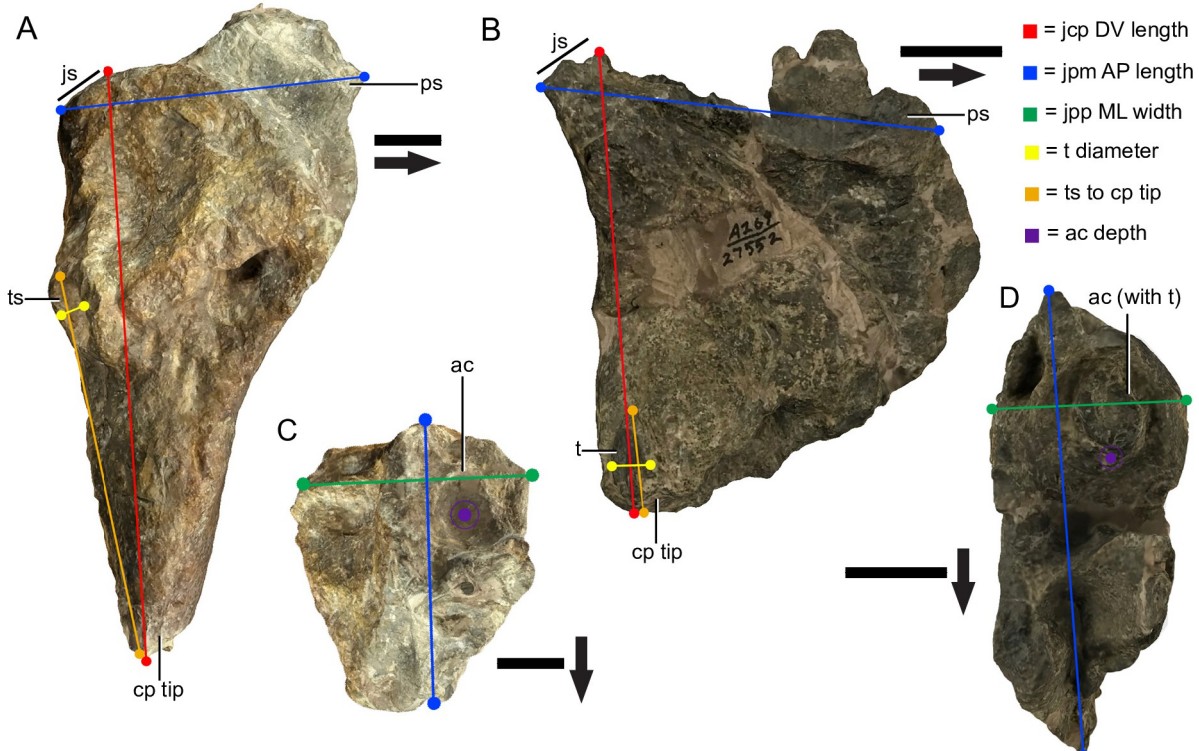

**Fig 2. Measurements taken from *Placerias* maxillae.** (A) Medial view of a left large-caniniform morph maxilla (UCMP 24935); (B) Medial view of a left small-caniniform morph maxilla (UCMP 27552); (C) Dorsal view of UCMP 24935; (D) Dorsal view of UCMP 27552. *Abbreviations*: *js*, jugal suture; *ps*, premaxillary suture; *ts*, tooth socket; *t*, tooth; *cp*, caniniform process; *ac*, alveolar cavity; *jcp* DV, jugal-caniniform process dorsoventral, *jpp* ML, jugal-pterygoid pit mediolateral; *jpm* AP, jugal-premaxillary anteroposterior. Arrows denote anterior direction. Scale bars equals 2 cm.

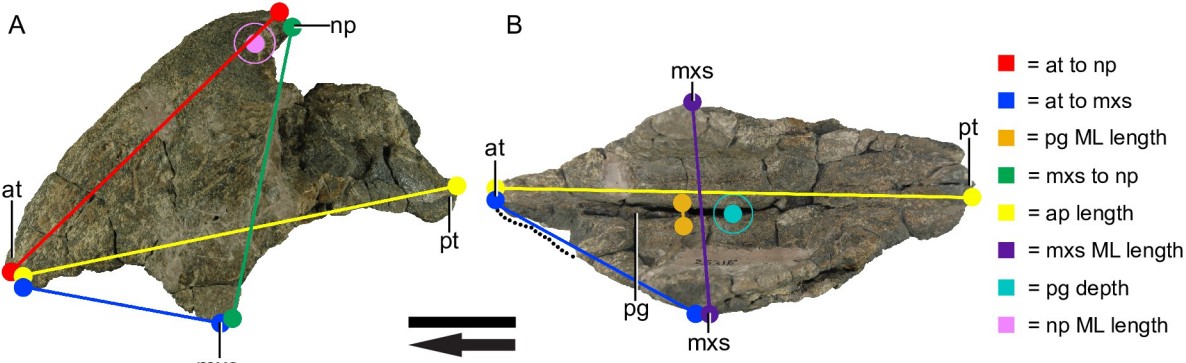

**Fig 3. Measurements taken from *Placerias* premaxillae.** UCMP 25316. (A) Left lateral view; (B) Ventral view; *Abbreviations*: *at*, anterior tip; *np*, nasal process; *mxs*, maxillary suture; *pg*, palatal groove; *ML*, mediolateral; *AP*, anteroposterior. Arrows denote anterior direction. Scale bar equals 5 cm.

maxillary sutures (mxs ML), depth of palatal groove (pg depth), and nasal process mediolateral length (np ML), (Fig 3).

The quadrates of *Placerias* have two condyles, the medial mandibular condyle (MMC) and the lateral mandibular condyle (LMC). For each condyle three measurements were recorded: the dorsoventral (DV), mediolateral (ML), and anteroposterior (AP) lengths, totaling six measurements (Fig 4).

For the fibulae, the following five measurements were recorded: distance from the tibial condyle to the femoral condyle (tc to fc), length of the distal condyle from tip to tip (dc tip to tip), distance from the tibial condyle to the distal condyle (tc to dc), the minimum dorsoventral midshaft diameter (ms DV), and the dorsoventral length of the proximal condyle (pc DV) (Fig 5).

All measurements were taken to the nearest millimeter using 15cm long calipers or, for longer lengths, 1m long measuring tape, (measurement error of 1 mm), and were $\log_{10}$ transformed prior to analysis.

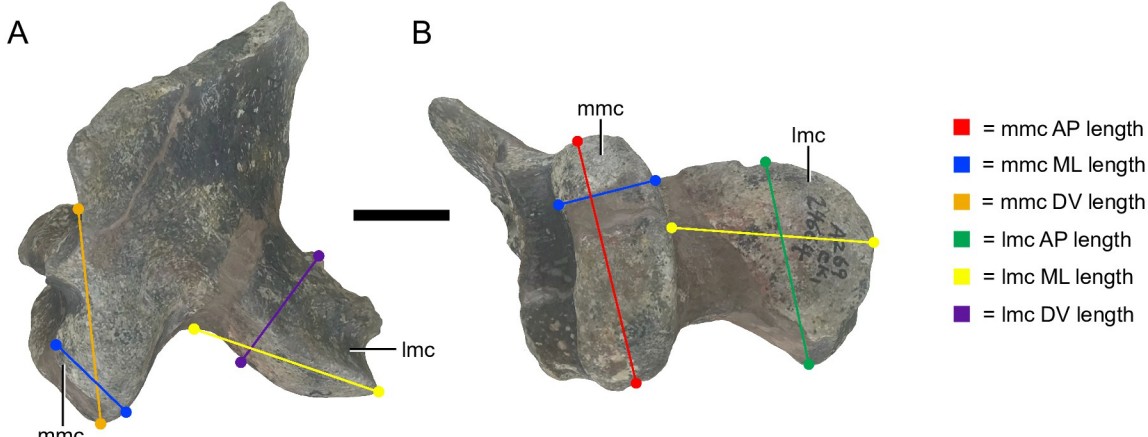

**Fig 4. Measurements taken from *Placerias* quadrates.** UCMP 24664, a right quadrate. (A) Posterior view; (B) Ventral view. *Abbreviations*: *mmc*, medial mandibular condyle; *lmc*, lateral mandibular condyle; *DV*, dorsoventral; *ML*, mediolateral; *AP*, anteroposterior. Scale bar equals 2 cm.

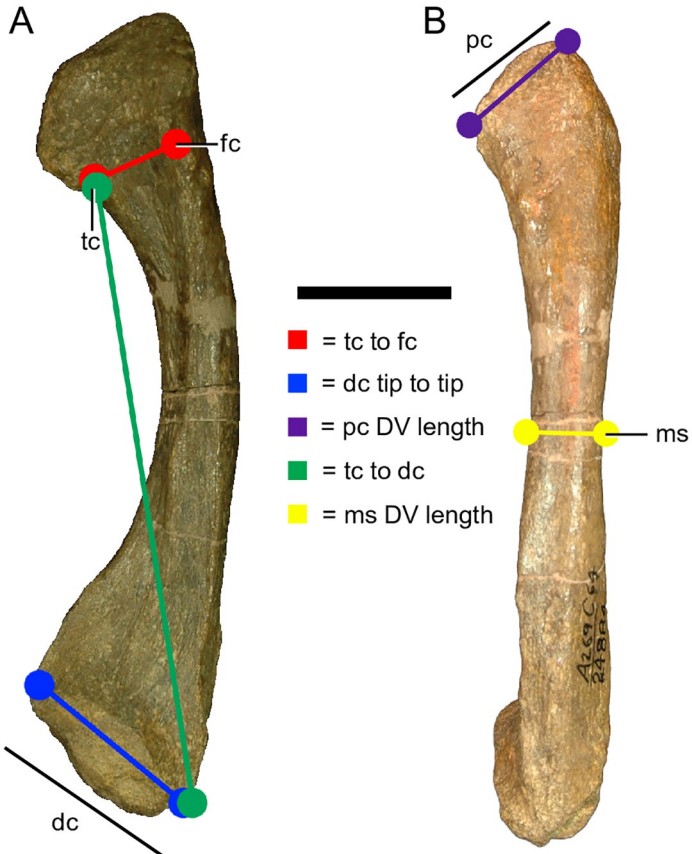

**Fig 5. Measurements taken from *Placerias* fibulae.** UCMP 24884, a right fibula. (A) Posterior view; (B) Lateral view. *Abbreviations*: *tc*, tibial condyle; *fc*, femoral condyle; *dc*, distal condyle; *pc*, proximal condyle; *ms*, midshaft; *DV*, dorsoventral; *ML*, mediolateral. Scale bar equals 5 cm.

## Data analysis

We used two approaches to quantitatively test the hypothesis of dimorphism in *Placerias* for each of the four elements we measured. First, we used a maximum likelihood approach to estimate parameters that best describe linear measurements by comparing the fit of a unimodal and a bimodal distribution with Akaike's Information Criterion corrected for small samples (AICc). We used a normal (Gaussian) distribution for the unimodal distribution, which requires the estimation of two parameters (mean and variance). For the bimodal distribution, we used a mixture of two normal distributions (Normal$_1$ and Normal$_2$), with a total of five parameters, the mean and variance of each normal ($\mu_1, \sigma_1^2, \mu_2, \sigma_2^2$) and a mixture parameter "*a*" that determines the relative contribution of each of the two normal distributions when they are combined. The parameter "*a*" is bound to vary between 0 and 1, where Normal$_1$ is multiplied by "*a*" and Normal$_2$ multiplied by "1-*a*". If "*a*" equals 0.5, then the data are best explained by two equal distributions, as would be expected from a 1:1 ratio of values from each distribution. This model fitting approach was repeated for each length measurement (a total of 25) on the four bones we sampled.

Second, we performed a principal component analysis (PCA) for each bone, with the variance of each measurement scaled so that measurements with larger values did not overly affect the results, to determine the relative position of each specimen in principal component space.

We used the new PCA coordinates of each specimen as input for the clustering analysis *k*-means, which finds the best way to separate the data into two groups, a number preselected in the clustering analysis. We used scatter plots to visually inspect the two groups detected by *k*-means and determine if they lack overlap, indicating they are likely separate morphotypes, or if they overlap, indicating that they cannot be clearly separated. All statistical calculations were performed in R, and scripts are available as supplementary material (S2 Table and S1 File).

For the measurements that were better explained by the bimodal distribution, graphs were used to visualize which specimens contributed most to the positions of $Normal_1$ and $Normal_2$. Note that the two distributions in the mixture function can sometimes have significant overlap, and that specimens in that region cannot be assigned to one group or the other. The bimodal model fitting and k-means clustering allowed for separation of specimens into groups (morphs). With this *a priori* knowledge we examined the potential correlation between sets of lengths, including those that were not distinct enough on their own to separate individuals into groups, by plotting them against each other.

Institutional Abbreviations: MNA, Museum of Northern Arizona, Flagstaff, AZ, USA; PEFO, Petrified Forest National Park, AZ, USA; UCMP, University of California Museum of Paleontology, Berkeley, CA, USA.

## Results

### Maxillae

We found stronger support for a bimodal distribution than for a unimodal distribution in the 16 maxillae fully intact for jugal-caniniform process dorsoventral (jcp DV) length (Fig 6A, Table 1). This measurement best captures the length of the caniniform process, and as such this result corroborates Camp & Welles' [29] hypothesis of dimorphism for this trait in *Placerias*. The parameters that describe the bimodal distribution indicate an equal proportion of specimens contributing to each of the two distributions (parameter *a* = 0.5). The smaller distribution has a mean of 130.6mm (2.1159 in $\log_{10}$), and the larger has a mean of 237.66mm (2.37596 in $\log_{10}$). The standard deviations also differ, with the large-caniniform morph's being much greater (large-caniniform morph SD = 0.064, small-caniniform morph SD = 0.034), and the coefficient of variation in the large-caniniform morph distribution is about 1.7 times that of the small-caniniform morph (large-caniniform morph CV = 0.0274, small-caniniform morph CV = 0.0159), indicating a greater variance in the jcp DV lengths of large-caniniform morphs even after taking into account their larger size. Other measurements of the maxillae show mixed results. Similar to the caniniform process length, the jugal-pterygoid pit mediolateral (jpp ML) width of the maxillae was better described by a bimodal than a unimodal distribution for the 28 maxillae complete for this measurement (Fig 6B, Table 1). However, in the 23 maxillae with fully intact jugal-premaxillary anteroposterior (jpm AP) lengths, there is strong statistical support for a unimodal distribution (Fig 6C, Table 1).

The clustering analysis k-means also detects two distinct groups of specimens, which can be seen when the jcp DV, jpp ML, and jpm PAP lengths are plotted in a PCA (Fig 7). The distribution of specimens between the two clusters corresponds exactly with their classification based on jcp DV length, and they are mostly separated along the second principal component (Dim2), indicating this length alone is sufficient alone to determine whether a specimen is a small-caniniform (hereafter referred to as "small") or large-caniniform (hereafter referred to as "large") morph.

In the 16 specimens with both the jcp DV length and jpp ML widths intact, the small and large morph are clearly separated along both of these lengths (Fig 8A). There is also a strong positive correlation between jcp DV length and jpp ML width in the large morph ($r^2$ = 0.771,

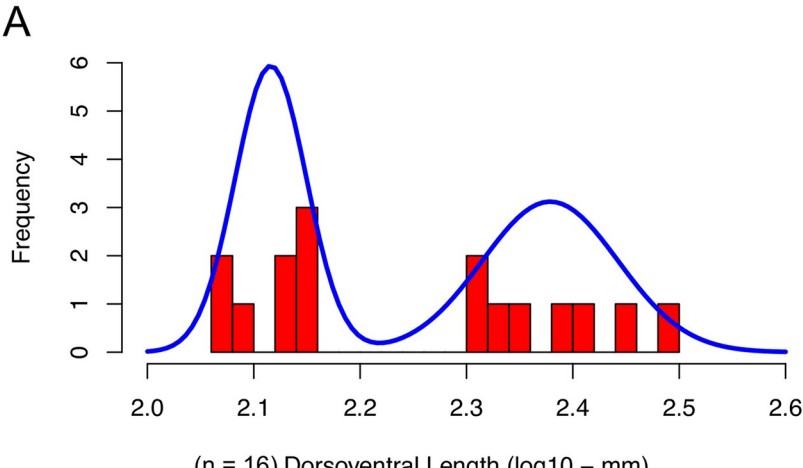

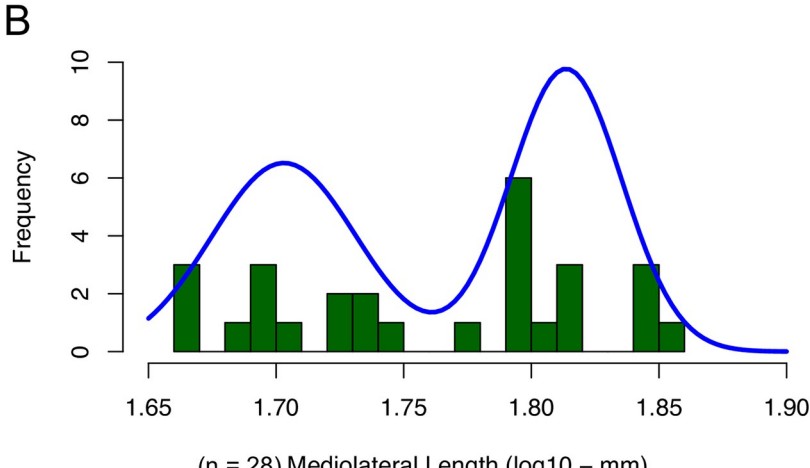

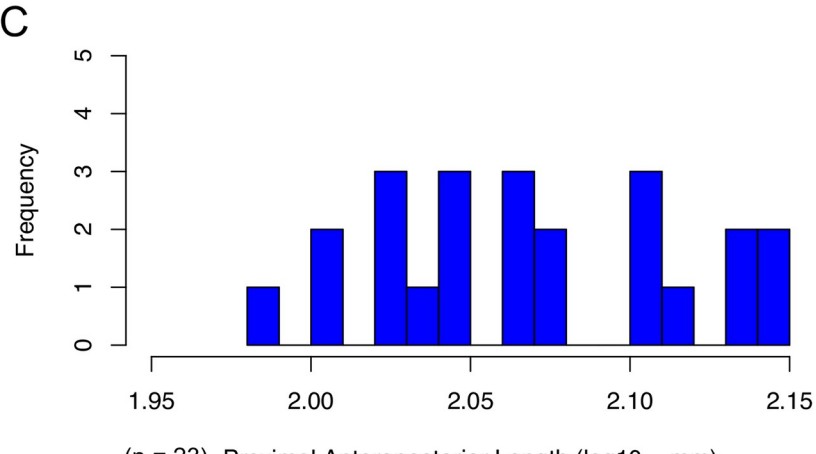

**Fig 6. Histograms showing distributions of lengths in *Placerias* maxillae.** (A) Jugal-caniniform process dorsoventral (jcp DV) lengths of 16 maxillae fully intact along that length, with a curve showing supported bimodal distribution. (B) Jugal-pterygoid pit mediolateral (jpp ML) widths of 28 maxillae fully intact along that width, with a curve showing supported bimodal distribution. (C) Jugal-premaxillary anteroposterior (jpm AP) length of 23 maxillae fully intact along that length, with a supported unimodal distribution.

**Table 1. Log-likelihoods and AICc comparisons of bimodal and unimodal models for each measurement from different bone elements of *Placerias*.** See text and figures for measurement acronyms. 'n' is the number of specimens measured; dAICc is AICc difference for the model with the lowest AICc.

| Bone | Measurement | n | Unimodal (2 parameters) | | | Bimodal (5 parameters) | | |
|------|-------------|---|------|------------|----------------|------|------------|----------------|
| | | | dAICc | AICc weight | log Likelihood | dAICc | AICc weight | log Likelihood |
| Maxilla | jcp DV | 16 | 2.44 | 0.23 | 8.6 | 0 | **0.77** | 15.4 |
| Maxilla | jpm AP | 23 | 0 | **0.89** | 38.0 | 4.1 | 0.11 | 40.4 |
| Maxilla | jpp ML | 28 | 4.7 | 0.09 | 38.8 | 0 | **0.91** | 45.2 |
| Maxilla | td | 19 | 0 | **0.53** | 25.7 | 0.2 | 0.47 | 30.5 |
| Maxilla | ad | 20 | 0 | **0.95** | 11.8 | 6 | 0.05 | 13.7 |
| Maxilla | ttc | 14 | 0 | **0.99** | -2.6 | 8.5 | 0.01 | -0.7 |
| Premaxilla | at to np | 7 | 0 | **>0.999** | 10.6 | 57.0 | <0.001 | 13.6 |
| Premaxilla | at to mxs | 8 | 0 | **>0.999** | 8.7 | 31.3 | <0.001 | 9.9 |
| Premaxilla | pg ML | 7 | 0 | **>0.999** | 6.1 | 62.4 | <0.001 | 6.3 |
| Premaxilla | mxs to np | 9 | 0 | **>0.999** | 11.4 | 19.4 | <0.001 | 13.7 |
| Premaxilla | AP | 6 | 0 | **>0.999** | 9.7 | Inf | <0.001 | 16.1 |
| Premaxilla | mxs ML | 7 | 0 | **>0.999** | 9.5 | 58.6 | <0.001 | 11.7 |
| Premaxilla | pg depth | 8 | 0 | **>0.999** | 8.5 | 29.6 | <0.001 | 10.5 |
| Premaxilla | np ML | 8 | 0 | **>0.999** | 10.7 | 28.2 | <0.001 | 13.4 |
| Quadrate | mmc AP | 40 | 0 | **0.9** | 64.2 | 4.4 | 0.1 | 65.7 |
| Quadrate | mmc ML | 42 | 0 | **0.86** | 54.5 | 3.6 | 0.14 | 56.4 |
| Quadrate | mmc DV | 41 | 0 | **0.51** | 63.8 | 0.1 | 0.49 | 67.5 |
| Quadrate | lmc AP | 31 | 0 | **0.69** | 40.8 | 1.6 | 0.31 | 44 |
| Quadrate | lmc ML | 29 | 0 | **0.96** | 40.9 | 6.6 | 0.04 | 41.7 |
| Quadrate | lmc DV | 29 | 0 | **0.85** | 38.2 | 3.5 | 0.15 | 40.5 |
| Fibula | tc to fc | 14 | 0 | **0.98** | 26 | 7.6 | 0.02 | 28.4 |
| Fibula | dc tip to tip | 12 | 0 | **0.98** | 19.2 | 7.8 | 0.02 | 22.6 |
| Fibula | pc DV | 14 | 0 | **0.97** | 21.3 | 6.8 | 0.03 | 24.1 |
| Fibula | tc to dc | 8 | 0 | **>0.999** | 18.3 | 28 | <0.001 | 21.1 |
| Fibula | ms DV | 12 | 0 | **0.82** | 21.1 | 3.1 | 0.18 | 26.8 |

$p = 0.002$, $n = 8$), but not in the small one ($r^2 = -0.133$, $p = 0.685$, $n = 7$). This likely reflects the caniniform becoming rounder in cross section in the large morph with increasing jcp DV length. This further distinguishes the large morph from the small one, which is more wedge-shaped in cross section, and further substantiates the differences between the two morphs. For the jcp DV and jpm AP lengths in the 14 specimens with both of these lengths intact, the distributions of jpm AP length in each morph differ, but overlap (Fig 8B). There is also a positive correlation between these lengths in the large morph ($r^2 = 0.505$, $p = 0.029$, $n = 8$), but not in the small one ($r^2 = 0.28$, $p = 0.1275$, $n = 7$).

The small morph maxilla of *Placerias* is relatively consistent in proportion between individuals (Fig 9A). Generally, jcp DV and jpm PAP length are much closer to equal than in the large morph, with the jpp ML width being larger relative to jpm AP length. This results in an overall "slimmer" form in the small morph, with a single anterior edge, and a posterior "wall" with medial and lateral edges, leading to the wedge-like dorsoventral cross-sectional shape. The large morph maxilla of *Placerias* is less consistent in proportion between individuals, but is still distinct from the small morph (Fig 9B). The jcp DV length is always longer than jpm AP length, in some individuals by over 2 times. As jcp DV length and jpp ML width increase, the anterior surface of the maxilla, particularly the portion ventral to the tooth, becomes less of an edge, becoming round in cross section. The caniniform in the larger morph extends ventrally beyond the rest of the skull (Fig 10).

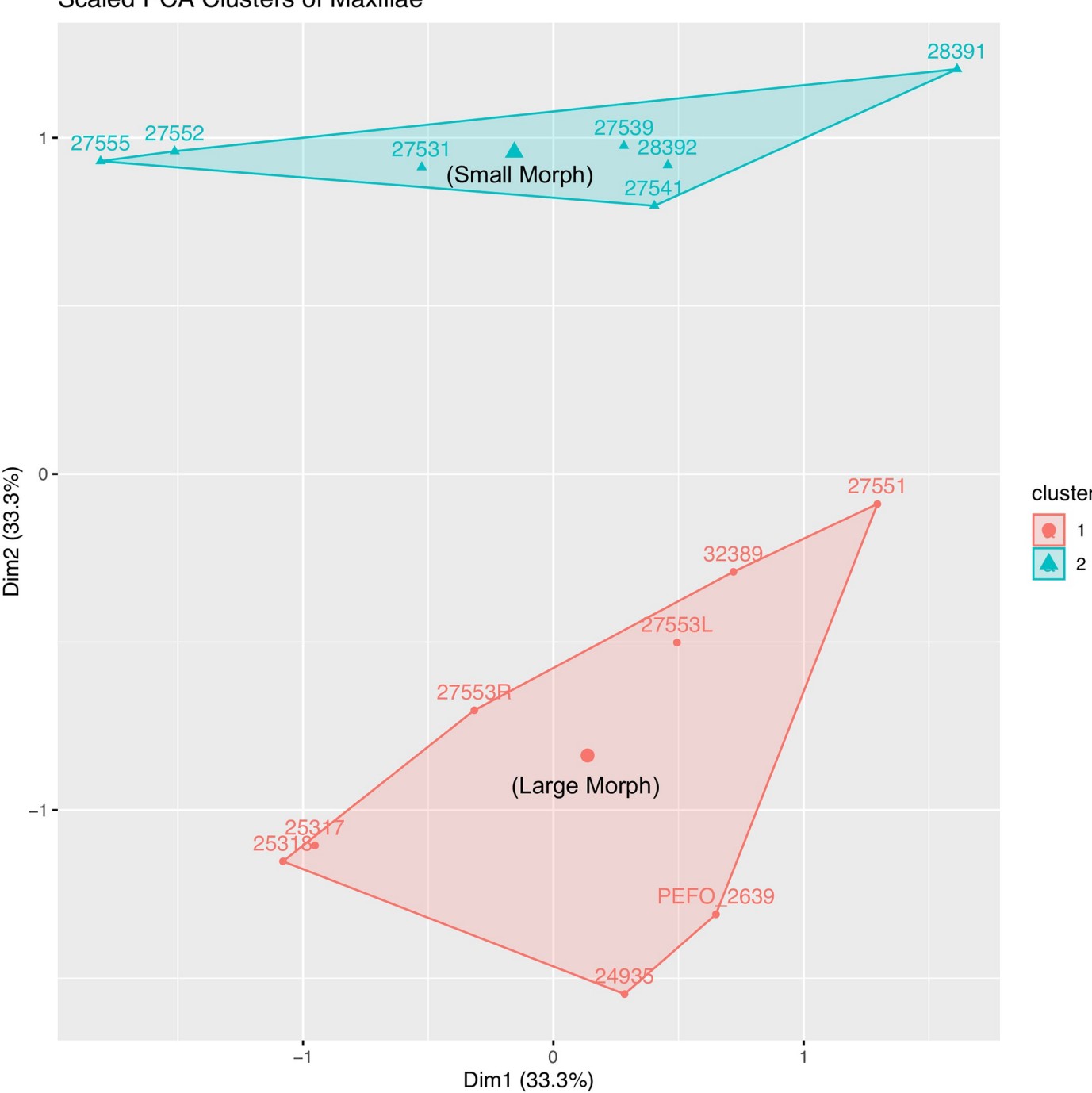

**Fig 7. PCA plot of 15 maxillae from the *Placerias* Quarry population, based on the jugal-caniniform process dorsoventral (jcp DV) length, jugal-pterygoid pit mediolateral (ML) width, and jugal-premaxillary anteroposterior (jpm AP) length.** Clusters from k-means align with assignment of large and small morphs based on dorsoventral length.

In large-caniniform morphs, the tooth is distant from the ventral edge of the maxilla due to the increased caniniform length, but relative to other features, such as the anterior pit and premaxillary and jugal sutures, it is in roughly the same location as in the small-caniniform morph. There is a proportional increase in jcp DV length in the large morph across the entire

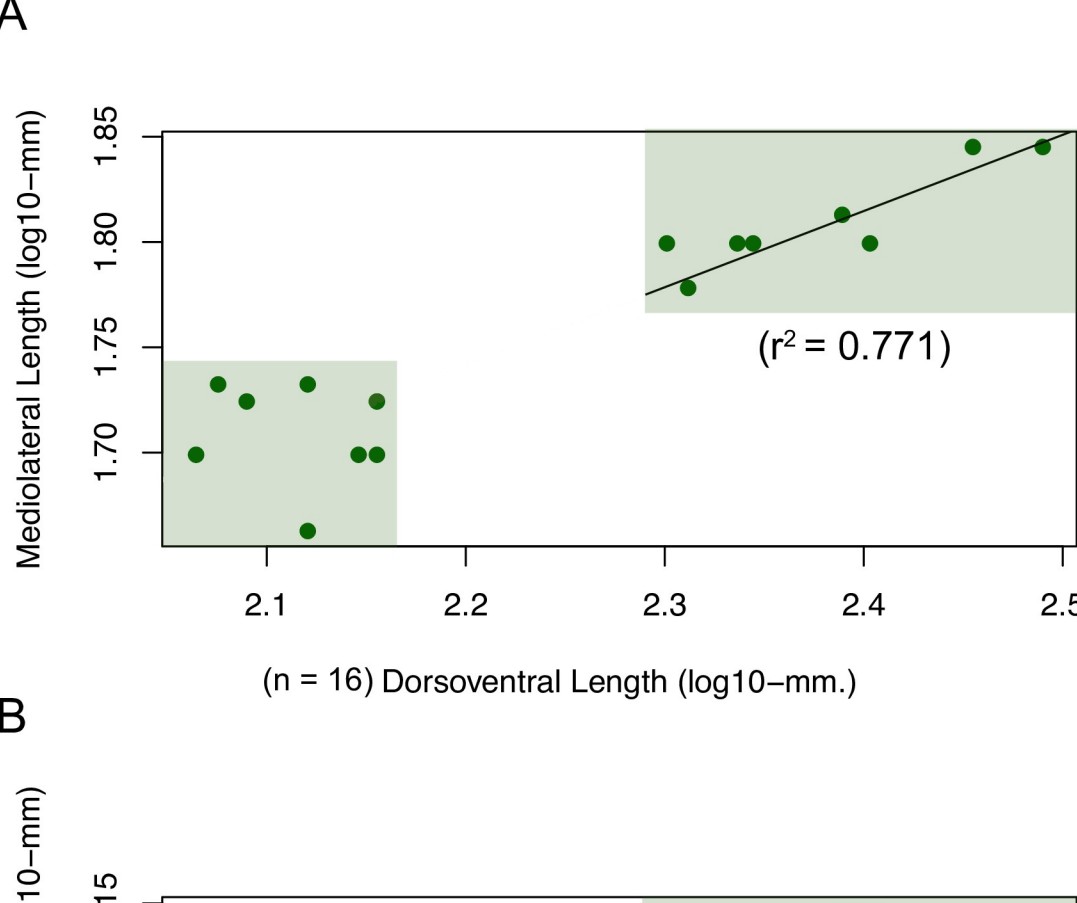

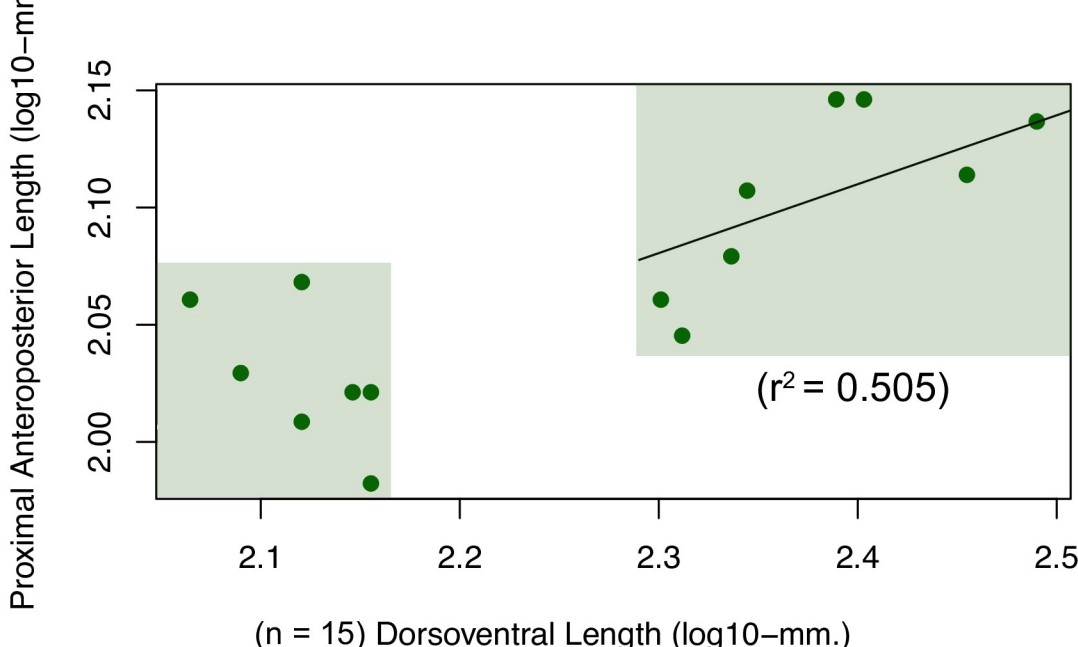

**Fig 8. Scatter plots showing relationships between lengths in *Placerias* maxillae.** (A) jugal-caniniform process dorsoventral (jcp DV) length and jugal-pterygoid pit mediolateral (jpp ML) width; (B) jcp DV and jugal-premaxillary anteroposterior (jpm AP) lengths. The $r^2$ values show the strength of correlation between lengths in the large morph; in the small morph, neither pair of lengths was significantly correlated.

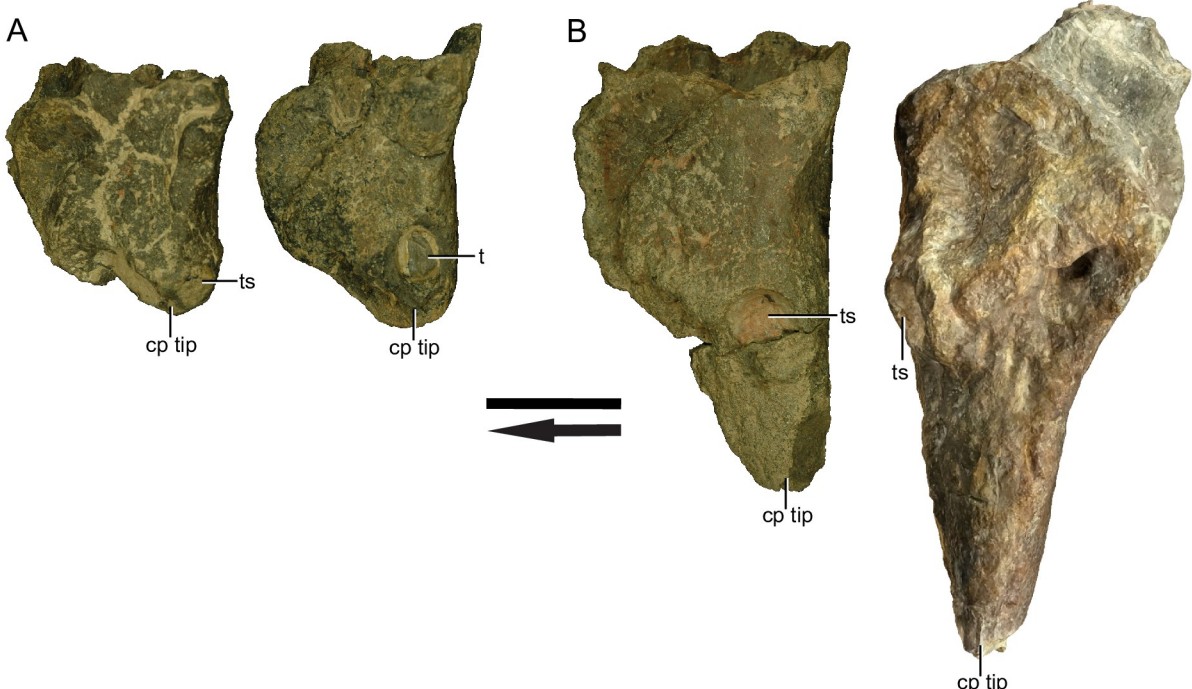

**Fig 9. Medial views of exemplar "small" (A) (from left to right: UCMP 28392 and 27541) and "large" (B) (from left to right: UCMP 27551 and 24935) morphs.** Note the greater variance within the large morph. Classification of morphs is based on the difference in proportion between jugal-caniniform process dorsoventral (jcp DV) length and jugal-pterygoid pit mediolateral (jpp ML) width, compared to jugal-premaxillary anteroposterior (jpm AP) length. *Abbreviations*: *ts*, tooth socket; *t*, tooth; *cp*, caniniform process; Arrow denotes anterior direction. Scale bar equals 5 cm.

maxilla, not just in the area between the caniniform and the tooth socket, but it appears that most of the difference in jcp DV length between morphs is ventral to the tooth socket. This is also indicated by the strong positive correlation between longer tooth socket-to-caniniform tip length and overall jcp DV length across both morphs ($r^2 = 0.905$, $p = 1.058 * 10^{-7}$, $n = 14$). Taking the morphs by themselves, this correlation is also present in the large morph ($r^2 = 0.870$, $p = 0.0004$, $n = 8$), but not the small morph ($r^2 = -0.196$, $p = 0.619$, $n = 6$). Given this and the supported bimodality in jcp DV length, it would be expected for the distribution of lengths from tooth socket to caniniform tip taken by itself to be bimodal. However, model comparison indicated that this length is better described by a single unimodal distribution than a bimodal one (ttc in Table 1).

There is more relative variation in ttc length than in jcp DV length, but the lack of support for bimodality may result from the amount of ttc length variation within each morph being close to the amount of variation between them. While the small-caniniform morph maxillae all have much smaller ttc lengths than any large-caniniform morph maxillae, the variance within each morph, particularly the large-caniniform morph, significantly impacts the overall distribution. The relative size of gaps in the expected range of ttc length within each morph are as large as the gap between the ranges of the small and large morphs, which results in the distribution having more than two clear clusters, and not being recognized as bimodal. These gaps in the expected ranges are likely the result of the relatively small sample size. With this small of a sample, the AIC comparison favors the simpler unimodal model with fewer parameters, and potential dimorphism has not been recognized.

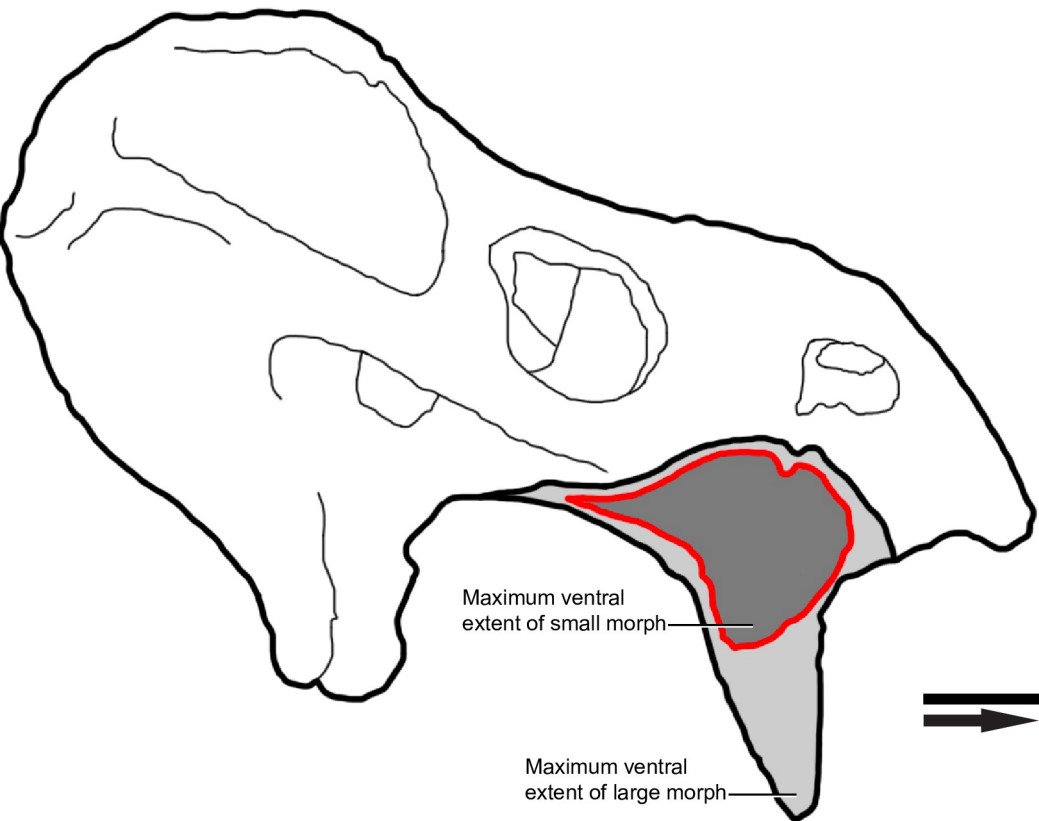

**Fig 10. An interpretive drawing of a *Placerias* composite skull (UCMP 137369).** Differing gray values show the different morphs of maxillae placed in the context of the rest of the skull. Note that the skull is scaled to the large morph, and the small morph likely was associated with a slightly smaller absolute skull size. Arrow denotes anterior direction. Scale bar equals 5 cm.

The depth of the proximal alveolar cavity (extending dorsally towards the inside of the skull) in the maxillae depends on the morph, with large morph maxillae having greater cavity depth. There is a weak but statistically significant ($p < 0.05$) correlation between alveolar cavity depth and both jcp DV length ($r^2 = 0.3025$) and jpp ML width ($r^2 = 0.5342$), traits shown to be dimorphic, but not proximal anteroposterior length, which lacks strong evidence for dimorphism. While alveolar depth itself was not shown to have a high log-likelihood of bimodal distribution in the 20 maxillae with the cavity intact (ad in Table 1), it correlates more strongly with the dimorphic caniniform traits than with absolute maxilla size.

## Tooth characters

Out of 32 maxilla which were intact enough to determine presence or absence of a tooth socket, four (UCMP 28391, 27547, 27538, PEFO 2639) lack any sign of tooth eruption or socket (Fig 11A). The other 28 maxillae all have a visible tooth socket present, indicating that the tooth erupted, though only 20 have the tooth present in the socket, with the teeth of the other 8 likely having been dissociated from their sockets post-mortem. In four different individuals (UCMP 25319, 27552, 27555, 32389) the tooth erupts out through both the distal and proximal end of the alveolus, projecting into the alveolar cavity of the skull (Fig 11B). Putative pathologies were also observed. Of particular note is specimen UCMP 27550, in which the

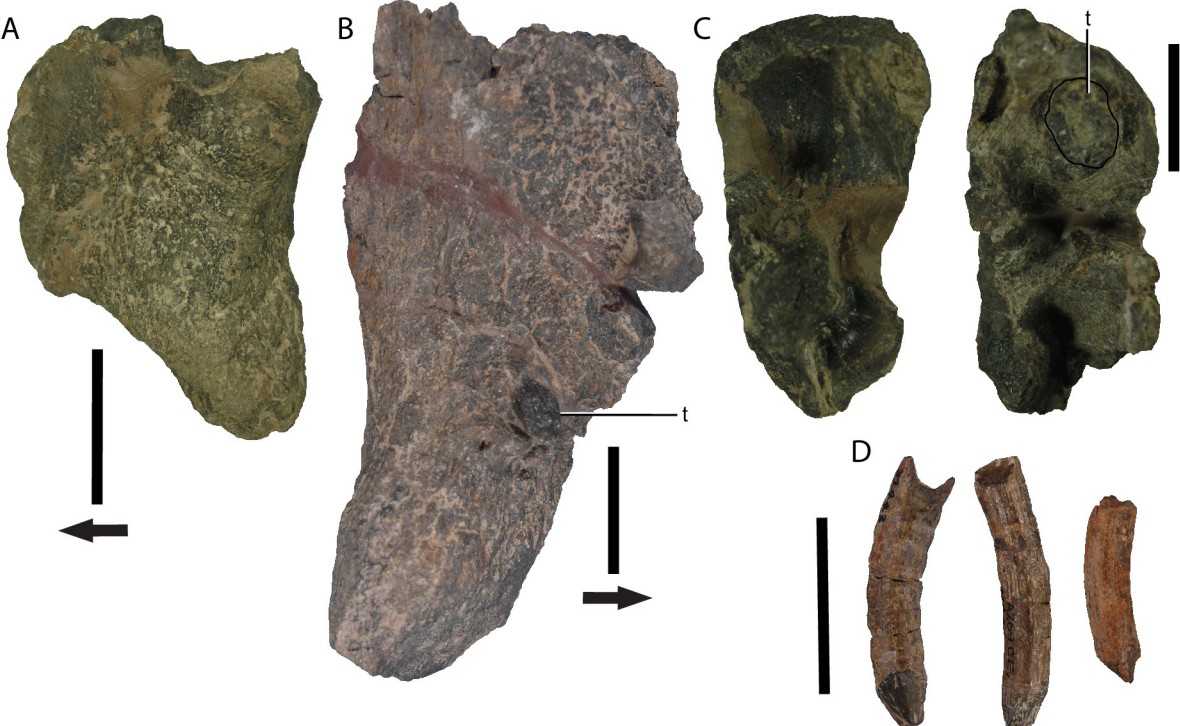

**Fig 11. *Placerias* maxillae and disassociated teeth.** (A) Medial view of a right small morph maxilla (UCMP 28391), with no sign of tooth eruption or tooth socket; (B) Medial view of a left large morph maxilla (UCMP 27550) with the tooth erupting near the anterior side of the maxilla instead of the typical eruption site near the posterior, and with abcessed surrounding bone. (C) Dorsal views of two right small morph maxillae (from left to right: UCMP 28391, UCMP 27552) with UCMP 27552 showing the uncommon condition of the tooth (circled) erupting through the proximal end of the alveolus; (D) Three disassociated teeth (from left to right: UCMP 32443, UCMP 32444, UCMP 24915) showing variance in the shape and surface texture of the root and crown, including differing extents of circumferential diameter oscillation ("ribbing") (left), of longitudinal grooves (right) running along the length of the root, and both of these features in combination (middle), along with more conical (right) and obliquely sheared (left) crown shapes. *Abbreviations*: t, tooth. Arrows denote anterior direction. Scale bars equal 5 cm.

tooth erupts out near the anterior edge of the maxilla instead of the posterior, and is surrounded by bone that appears to be abscessed (Fig 11C).

Disassociated teeth, which were found separated from maxillae and have fully exposed roots, have highly variable root morphology. In seven teeth, the tooth root shafts show variability in two main traits: the presence of longitudinal grooves running down the shaft, and the repeated oscillation of tooth diameter along the shaft (Fig 11D). Some teeth only show the oscillations (e.g. UCMP 32443), some show only the grooves (e.g. UCMP 24915), and some show both (e.g. UCMP 32444) (Fig 11D). These patterns suggest differences in growth between individuals, possibly related to differing rates of growth over the course of ontogeny. Irregular growth in dicynodont teeth has also been linked to developmental stress from sources like drought conditions [41]. The crown shape also varies by individual, with some (e.g. UCMP 24915) having a more pointed, cone-like tip, and others (e.g. UCMP 32444) having a single worn down edge, suggesting different levels of tooth wear between individuals. Overall, the differences in growth patterns and presence of pathologies in individuals that all reached adult sizes provides evidence that the tooth being present or in a particular position was not necessary for individual survival. This combined with the proportionately small size of the tooth, provide evidence that it likely evolved neutrally, with little pressure to maintain functionality, and was essentially vestigial.

Though Camp and Welles [29] speculated that tooth diameter and caniniform morphotype are correlated, we found no evidence for this. In the 19 measured maxillae with intact teeth, there is roughly equal support for a unimodal and a bimodal distribution (td in Table 1) of tooth diameter, with AICc weights of 0.53 and 0.47, respectively. However, tooth diameter does not correspond to the caniniform dimorphism, as some specimens with larger caniniforms have smaller tusks, and vice-versa. Further, the distributions are not distinct enough to separate teeth into unambiguous morphs as can be done for the caniniform, and tooth size is not correlated to dorsoventral or mediolateral length. Whether or not teeth are in fact dimorphic, evidence presented here for the tooth's lack of functionality and the relatively small range of their diameters (13–19mm) imply that any possible dimorphism had little functional significance.

## Premaxillae

In the nine premaxillae measured, there was no strong evidence of dimorphism in proportions. Of the eight lengths measured, none had strongly bimodal distributions. The small sample size makes it difficult to confidently assess the log-likelihoods of bimodality, but two morphs of premaxillae are also not apparent visually. The distributions of two traits, the distance from the anterior tip to the nasal process and the anteroposterior length, were only narrowly supported as normal (Table 1), with the AICc difference between distribution models in these traits not being sufficient to confidently support either a bimodal or unimodal distribution (Fig 12A), but all other measurements were strongly supported as having unimodal distributions.

Clustering the data on a PCA using k-means shows the lack of dimorphism in premaxillae when all measured traits are included (Fig 12B). However, removing the mediolateral width and the depth of the palatal groove (pg ML and pg depth in Table 1) from the PCA leads to two somewhat distinct k-means clusters (Fig 12C). This clustering is consistent with a statement made by Camp and Welles [29] that the "narrowest" premaxillae might belong to the small morph, and the "broadest" to the large morph. While our results do not explicitly support this interpretation, they do hint that the variance in the shape of the premaxilla may be in some way related to the morph of the maxilla, but it is also possible the differences are driven simply by variation in skull size, rather than sexual dimorphism.

## Quadrates

In the 44 quadrates measured, none of the six measured lengths had distributions that were strongly bimodal (Table 1). This is particularly important given the relatively large sample size for this element. However, while not statistically strongly supported as bimodal, distributions of three of the six lengths measured, all associated with the medial mandibular condyle (MMC), show visual signs of bimodality (Fig 13A–13C). This is reflected in the histograms of these traits, all of which are correlated to each other, having two visually apparent peaks. The MMC traits also separate into two distinct clusters based on k-means (Fig 13E). This points to the possibility that variance in MMC shape is best explained by two overlapping normal distributions that, taken by themselves, are not distinct enough to delineate morphs or statistically support a bimodal distribution. It is worth noting that the bimodal model had nearly equal likelihood than the unimodal model for the MMC dorsoventral (DV) length, despite that model receiving less AICc support after considering that it has higher number fitted parameters.

The distributions of the other lengths measured, namely those of the lateral mandibular condyle (LMC), were all supported as unimodal, and when added to the MMC PCA make the

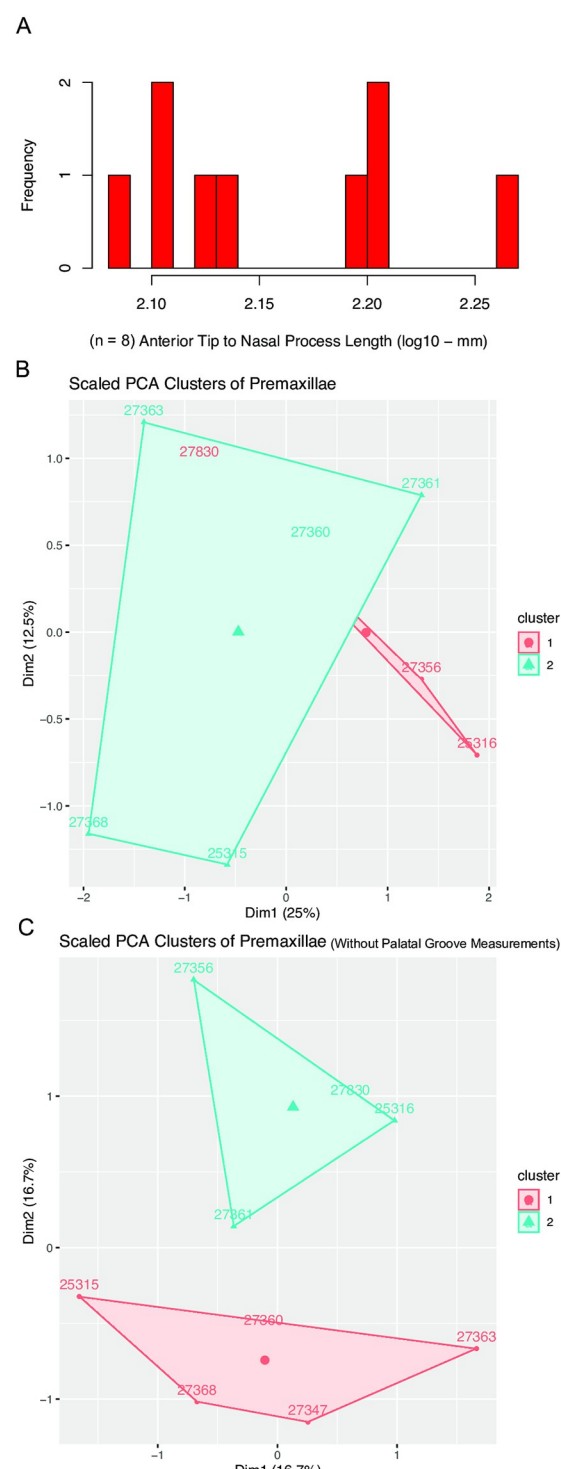

**Fig 12. Histogram of premaxillary anterior tip to nasal process lengths in the *Placerias* Quarry population, and PCA plot with k-means clusters.** (A) Distribution of the length from the anterior tip to nasal process, not showing high log-likelihood of either bimodality or unimodality, but having two peaks; (B) PCA plot of all measured premaxilla lengths grouped into 2 k-means clusters with high overlap, showing poor evidence for dimorphism. (C) PCA plot of measured premaxillae lengths excluding the palatal groove measurements, showing more distinct k-means clusters.

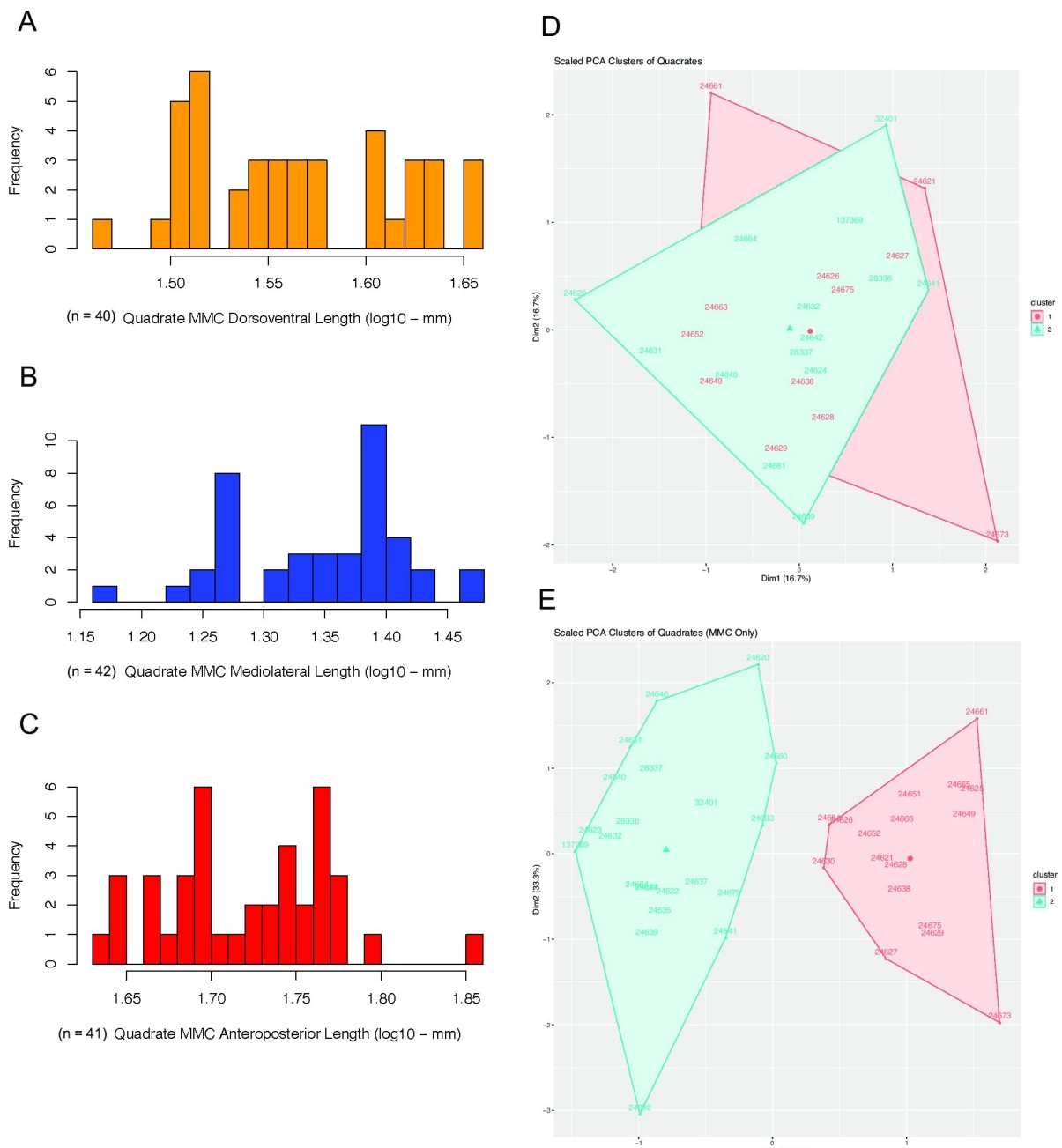

**Fig 13. Histograms of lengths in the medial mandibular condyle (MMC) of quadrates in the *Placerias* Quarry population, and PCA plots with k-means clusters showing potential dimorphism in the MMC.** (A) Distribution of MMC dorsoventral length, without a high log-likelihood of either bimodality or unimodality; (B) Distribution of MMC mediolateral length, showing possible visual bimodality, but with a high log-likelihood of unimodality; (C) Distribution of MMC anteroposterior length, also with a high log-likelihood of unimodality; (D) PCA plot of all measured quadrate lengths from the MMC and lateral mandibular condyle (LMC) grouped into 2 k-means clusters with high overlap, showing poor evidence for dimorphism; (E) PCA plot of measured quadrate MMC lengths, grouped into 2 k-means clusters with little overlap, showing potential evidence for dimorphism.

k-means clusters no longer distinct (Fig 13D). This suggests that there are two modules for the quadrate, the LMC and the MMC. While the data showed weak suggestions of bimodality on the MMC, which could be correlated with bimodality in the maxilla, the LMC does not show any indication of bimodality. The LMC pattern prevails when all quadrate measurements were

analyzed together in one PCA and k-means analysis. Thus, the potential dimorphism in the quadrate could be in the relative size of the MMC, rather than overall absolute size of the quadrate. However, the MMC was also represented in more specimens than the LMC, so this difference in apparent dimorphism could be the result of the different sample sizes for these portions of the quadrate. While there is no significant statistical support for dimorphism in the quadrate in *Placerias*, our data raise the possibility that the relative size of the MMC in the quadrate might be correlated to caniniform morph if associated with the maxilla.

## Fibulae

In the 17 fibulae measured, there is not strong evidence for proportional dimorphism. None of the five traits measured had strong support for a bimodal distribution (Table 1). Two traits, the length from the tibial condyle to the distal condyle (tc to dc in Table 1) and the dorsoventral midshaft diameter (ms DV in Table 1), have distributions with two visually apparent peaks (Fig 14A and 14B), but these lengths are represented by particularly low sample sizes; just eight fibulae were complete enough to measure tibial condyle to distal condyle length, and just 12 were complete enough to measure dorsoventral midshaft diameter. In both of these measurements, two fibulae are separated from the rest, these being the largest fibulae for all other measured lengths. These two fibulae also form a separate cluster based on k-means (Fig 14C), however when the tibial condyle to distal condyle length is removed the clusters are no longer distinct (Fig 14D). Given this, and the fact that measured lengths are all generally positively correlated with each other, it is possible that this difference in proportion stems from a positively allometric relationship between diaphyseal size and epiphyseal size of the fibula.

In the 2 larger fibulae, tibial condyle to distal condyle length and dorsoventral midshaft diameter, lengths related to the size of the diaphysis, are proportionately larger compared to the other measurements, which are related to the size of the epiphyses, than in the smaller fibulae. This would imply that diaphysis size increases faster than epiphysis size as overall body size increases. This may better explain the variation in fibula proportion than sexual dimorphism, and it may be more sensible to interpret the distribution of tibial condyle to distal condyle length (Fig 14A), which is the main driver of separation of clusters based on k-means (Fig 14C), as a normal distribution with two outlier values, as is supported statistically, rather than a bimodal distribution. While proportional diaphysis size could potentially be correlated with caniniform morph, with the individuals that have the proportionately largest diaphyses being from the large morph, the individual variation within fibulae from the *Placerias* Quarry population does not show strong evidence of dimorphism.

## Reassessment of MNA.V.8464

Currently the best preserved articulated skull of *Placerias* comes from a site outside of the *Placerias* Quarry, in another Chinle locality near Cameron, AZ [31]. The specimen consists of a fully articulated right half of a skull, including the maxilla. However, the specimen is not nearly as well preserved as the *Placerias* Quarry fossils, having been substantially laterally compressed, and having few discernible suture lines. This has led to difficulty in interpreting it, including in its initial description [31], which was literally "turned upside down" in a later publication [22], with the ventral and dorsal sides reversed. Kammerer et al., [22] argue that the fossil "confirms the accuracy of the Camp and Welles/Cox reconstruction", but no further statements were made. Since that publication, the skull has undergone additional preparatory work, from ~2019 to 2021, making its morphology clearer. Though the premaxilla is not intact, the maxilla and its suture with the jugal are clear enough to be compared to the *Placerias* Quarry material.

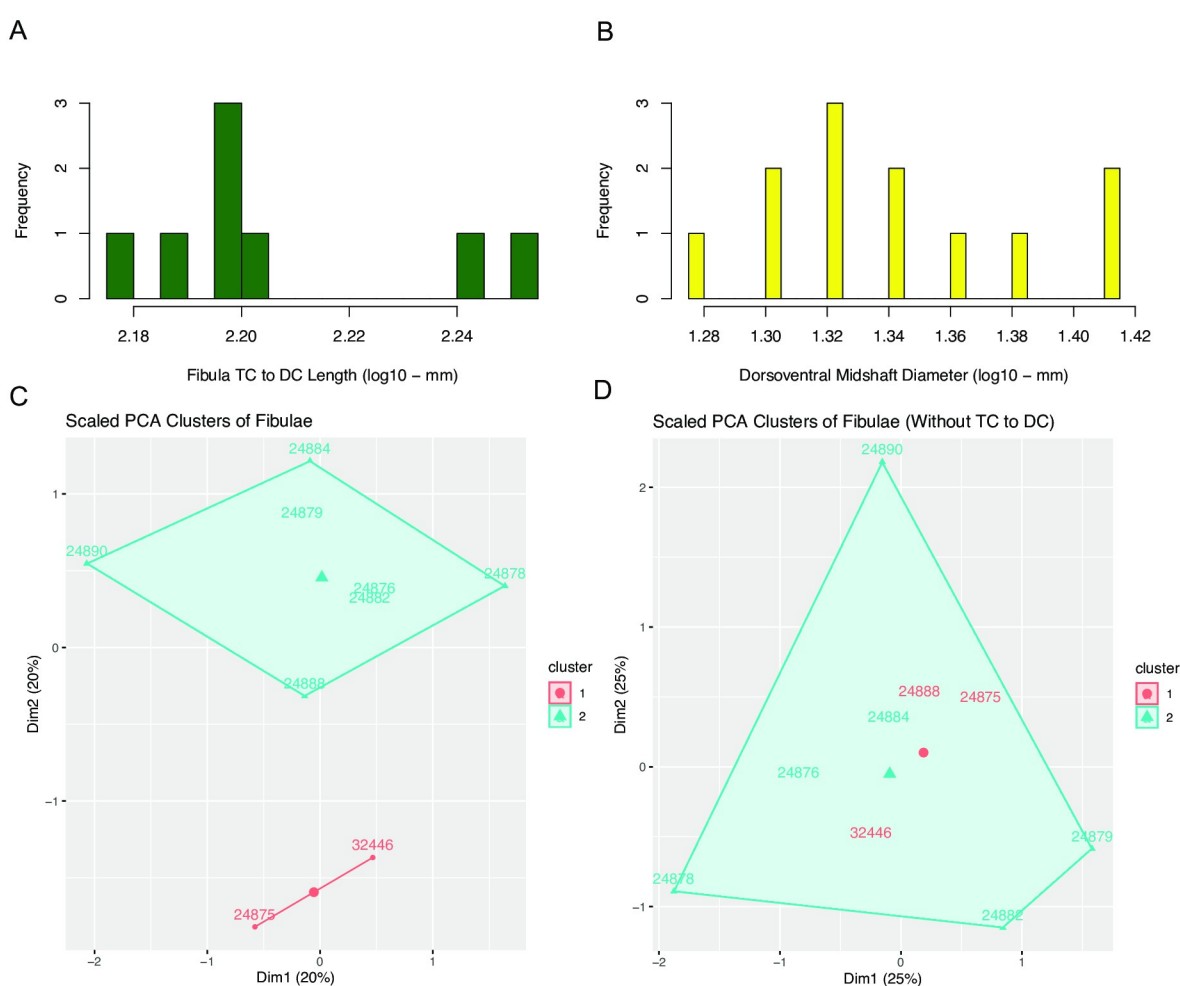

**Fig 14. Histogram of tibial condyle to distal Condyle (TC to DC) length of fibulae in the *Placerias* Quarry population, and PCA plots with k-means clusters showing variations in proportions of fibulae.** (A) Distribution of TC to DC lengths, with 2 larger fibulae visually separate from the rest of the sample, but statistically overwhelmingly supported as unimodal; (B) Distribution of dorsoventral midshaft diameters, with two visually apparent peaks, also supported as unimodal. (C) PCA plot of all measured traits, with k-means clusters showing separation of UCMP 24875 and 32446 from the rest of the sample; (D) PCA plot of all measured traits except for TC to DC length, showing relatively poor separation into distinct clusters through k-means.

The proportions of the maxilla from MNA.V.8464 fit well within the bounds of those of the small morphs from the *Placerias* Quarry population, with the jcp DV and jpm AP lengths being approximately equal. This, combined with the fact that other elements of the skull, such as the quadrate, are closer in size to the smaller end of individuals from *Placerias* Quarry, provides evidence that MNA.V.8464 likely represents a small morph individual of *Placerias*. This gives us a useful reference for the relative proportions of skull elements in the small morph, and can potentially be used for comparative purposes with the composite, as that reconstruction consistently uses some of the largest individual elements of the *Placerias* Quarry population, which more likely come from large morph individuals (Fig 15). This supports the interpretation that other than the significant elongation of the caniniform, the relative proportions of skull bones in *Placerias* remain similar between morphs, but that the two morphs may have differing distributions in overall size. Though there are some other proportional differences between MNA.V.8464 and the Cox [14] reconstruction (UCMP 137369), such as the

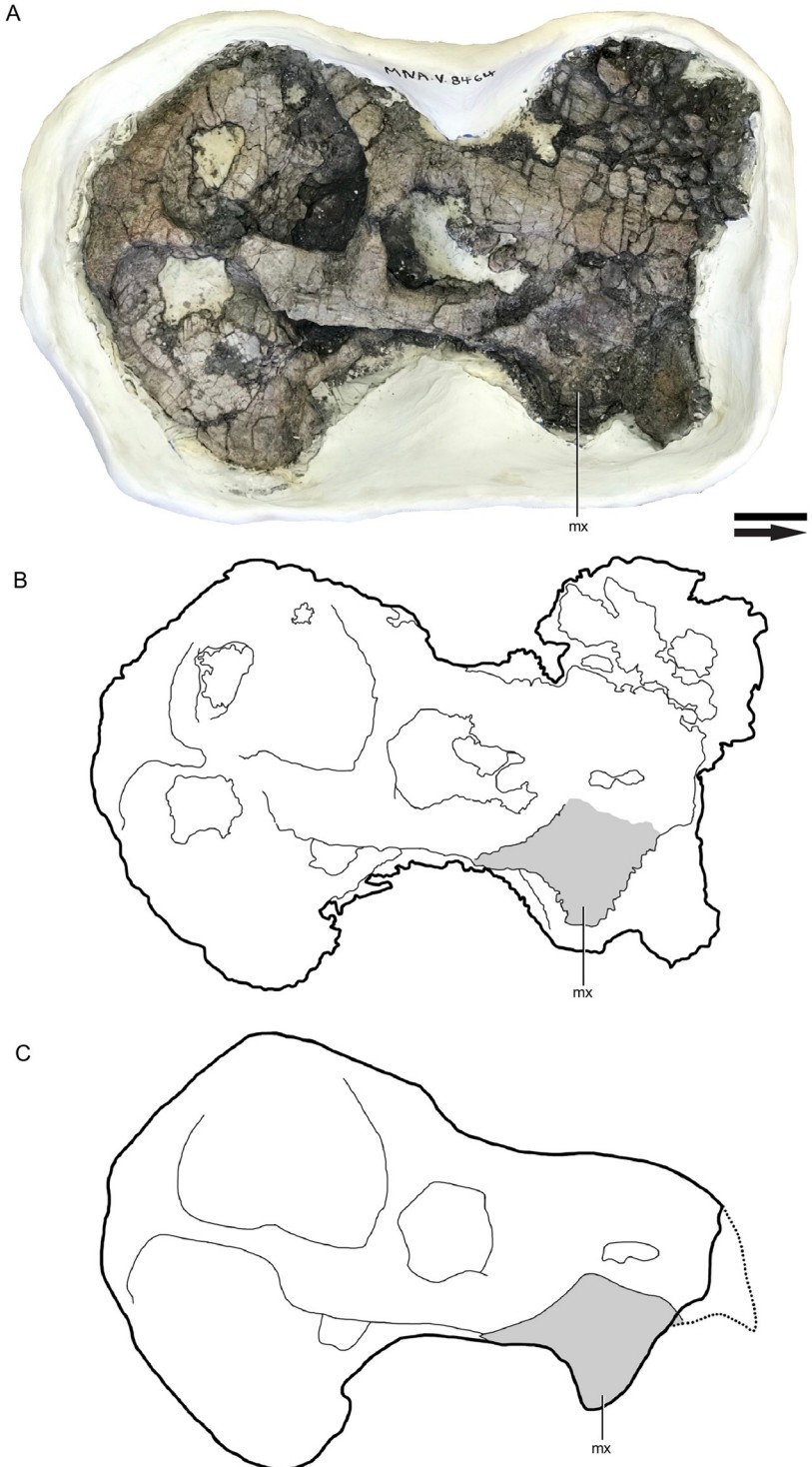

**Fig 15. Articulated *Placerias* skull (MNA.V.8464).** (A) Photo of right lateral view; (B), Interpretive drawing with maxilla highlighted in gray; (C) Reconstruction with anterior portion of premaxilla added in dotted line. *Abbreviations*: *mx*, maxilla. Scale bar equals 5 cm. Image courtesy of Museum of Northern Arizona.

anteroposteriorly shortened squamosal region and smaller pterygoid region in MNA.V.8464, these differences are not dramatic enough to be sure that they are not a product of deformation in MNA.V.8464, or inaccuracies in the reconstruction of UCMP 137369, or both.

## Discussion

In addition to sexual dimorphism, bimodality in size or proportion of bones being observed in a fossil assemblage can also result from changes through ontogeny, or anagenetic evolutionary change if the assemblage is time averaged. With this in mind, it is important to establish that the *Placerias* Quarry assemblage represents a population of *Placerias* that lived in the same place at the same time, and the individuals were approximately the same age, when trying to establish that observed conspecific differences constitute sexual dimorphism.

The *Placerias* Quarry is unusual compared to many sites from the Chinle Formation, both in the fauna represented and in the taphonomy of material preserved there. The most obvious distinction is the atypical abundance of *Placerias* itself, as well as a much smaller presence of amphibious and aquatic macrovertebrates relative to other Chinle Formation localities. The current understanding is that the *Placerias* Quarry was a seasonal floodplain, that the assemblage resulted from relatively rapid burial, and that the mechanism of death in *Placerias* was likely driven by seasonal drought and not extensive predation [34]. The lower level of *Placerias* Quarry, where the *Placerias* fossils are found, has been dated at 219.39 +/-0.12 Ma, based on U-Pb detrital zircon geochronology [42], placing it within the middle of the Norian. Low taphonomic grade across the specimens indicates they were exposed for short periods of time before burial, and very similar taphonomic grade across specimens is consistent with modern bone assemblages known to have formed on the order of decades [34]. Therefore the assemblage was likely composed of coeval animals, and though it is unknown if the *Placerias* lived in a group or herd, the preservation time of the assemblage was almost certainly insufficient for anagenesis to have occurred in the population; there was little to no time averaging.

Individual postcranial remains of *Placerias* from the site do vary in size (Fig 16), however the entire population appears to be composed entirely of either sub-adults or adults. This observation is supported by previous histological work performed on sampled limbs from the population [43], which concluded that there were no juveniles in the assemblage. Green et al. [43] further noted that even though there are bones of multiple sizes in their sample, the smallest having a midshaft diameter 57% the size of that of the largest individual, all bones sampled have extensive secondary remodeling, indicating maturity. Histological evidence shows that *Placerias* had a similar growth pattern to other Triassic dicynodonts, such as *Lystrosaurus* [43] and the placeriine *Moghreberia* [44], where periodic rapid osteogenesis was followed by extensive secondary remodeling and reduced growth of parallel-fibered bone later in ontogeny. In *Placerias*, some specimens that lack peripheral parallel-fibered bone are larger than others that have it, indicating later ontogeny in limb elements which substantially differ in size. This in combination with an inconsistent observed relationship between amount of lines of arrested growth, amount of remodeling, and limb midshaft diameter, suggests that *Placerias* varied in size to some degree as adults.

The size variation in adult *Placerias* is proposed by Green et al. [43] as being a possible result of either sexual dimorphism, developmental plasticity, or of multiple taxa being present in the sample. The possibility of multiple, very similar taxa being present at *Placerias* Quarry is difficult to disprove, however based on the lack of definable apomorphies that would otherwise separate *Placerias* into multiple taxa, the roughly 1:1 ratio of large morph to small morph individuals, the extremely sparse record of other North American dicynodont taxa, and the limited time involved in the formation of the fossil assemblage, the discrepancy in body sizes is much

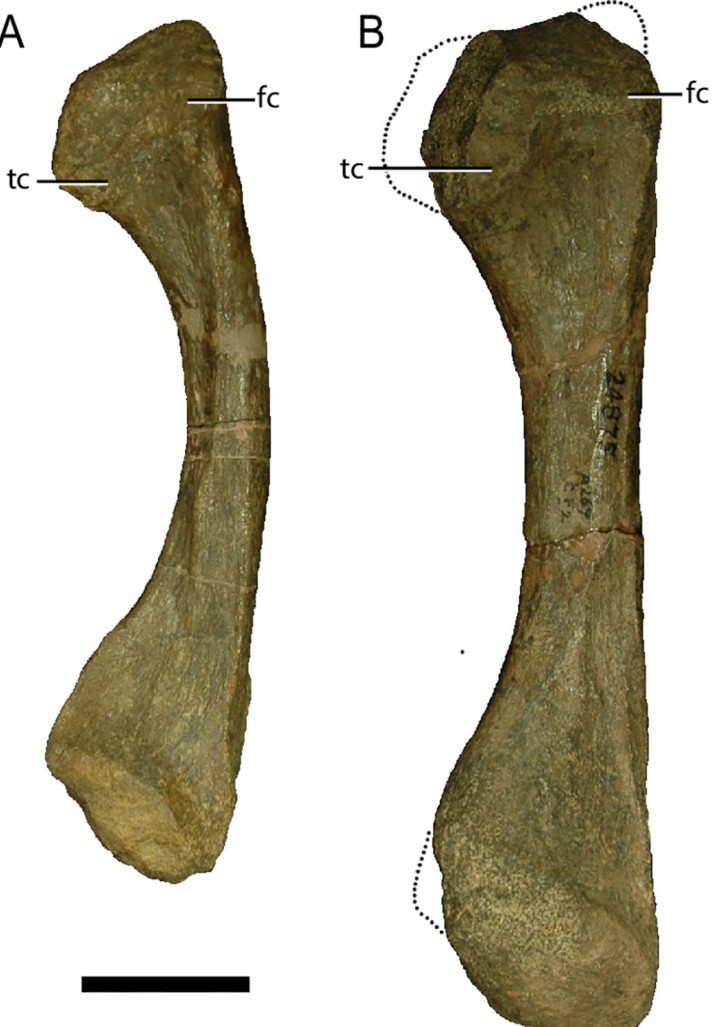

**Fig 16. Medial view comparison of *Placerias* fibulae.** (A) UCMP 24884 and (B) UCMP 24875 show a high degree of variance in individual sizes of elements from the *Placerias* Quarry population, despite histology providing evidence against young juveniles being in the population. *Abbreviations*: *tc*, tibial condyle; *fc*, femoral condyle. Scale bar equals 5 cm.

more easily explained by individual variation in a single taxon. Developmental plasticity could be at play, particularly due to the assumed stressor of drought conditions on the population based on the taphonomy of the site. That is, perhaps there was some time averaging of populations of subtly different ages, where the body size and ontogeny of some individuals was affected by drought, or some other factor. However, even if present, this does not rule out the likelihood that some portion of the observed differences was due to sexual size dimorphism (SSD).

Though the distribution of sizes in elements other than the maxillae are not strictly bimodal, elements with high sample sizes like the quadrate appear to have two overlapping normal distributions, which could possibly be separated into morphs when associated with a secondary sexual trait, as reflected in the correlation between jpm AP and jcp DV lengths in maxillae. Based on the statistical support for sexual dimorphism presented here, the most likely

explanation for the range of adult sizes in *Placerias* is SSD, consistent with some other synapsid megafauna. Considering the variation seen in the other measured elements, it is possible that cranial and postcranial element proportion varies with sex, but not to the extent that the morphs can be separated based on non-maxillary material alone. Though measuring other elements could show additional variation within the *Placerias* Quarry population, all of the other elements present (e.g. humeri, femora, etc.) are largely represented by incomplete fragments, and none have large enough complete sample sizes for stronger statistical analysis than those measured here.

## The caniniform as a secondary sexual trait

Based on the development of tusks, horns, and other sexually dimorphic weaponry and ornaments, along with differences in body size and robustness of cranial features in extant synapsids (i.e., mammals) [8], it seems likely that the small morph maxillae represent female individuals and the large morph males. This would also be consistent with the higher prevalence of intrasexual competition among males of mammals, and males being more frequently the sex that invests less in offspring than females. However, this cannot be determined for certain, and in any case not relevant to the claim that dimorphism is present in the population.

Though the most obvious point of difference between the large and small morph caniniform is their dorsoventral length, the variation in mediolateral and proximal anteroposterior lengths of the maxillae indicates there are other differences in the shape of the caniniform. There is strong statistical support for bimodality in mediolateral length of maxillae, and while there is overlap in the two distributions, as might be expected [4], which confounds unequivocal assignment of all specimens to one of the two distributions, there is complete agreement in the individuals grouped by caniniform length and the mediolateral length. Proximal anteroposterior length is less consistent with morph or dorsoventral length, but is still correlated. The maxilla connects to the rest of the skull along this length, making it a reasonable proxy for overall skull size in individuals. The unimodal distribution for this length indicates that although there is variation in absolute skull size among individuals, the distribution of this variation is different from that of the sexually dimorphic caniniform.

Sexual dimorphism in the caniniform has not been quantitatively reported in other dicynodonts. Other taxa speculated to have sexual dimorphic cranial features, such as *Aulocephalodon* and *Pelanomodon*, are placed in morphs based on the thickness of nasal, premaxillary, and prefrontal bosses, and rugose bone around these areas [12, 17]. *Placerias*, in contrast, has relatively thin, smooth nasals and prefrontals. These elements also show very little variation in the *Placerias* Quarry population, though their relatively small sample sizes could be partially responsible for this. These conditions, and the relative lack of bimodality seen in the premaxillae, imply the lack of a highly developed or dimorphic rostral boss in this species. Given that none of the elements that we measured other than the maxilla have strong statistical support for a bimodal distribution (dimorphism), it appears that the caniniform does indeed represent a secondary sexual feature; a trait that was selected for in a single sex for a function related to sexual competition.

In *Placerias*, though the tooth is still present in most individuals, it appears to have been functionally replaced by the caniniform process. In some other Stahleckeriids, the tooth is entirely missing (at least in all known individuals) [16], also likely replaced in function by the caniniform. The teeth of *Placerias* are underdeveloped, measuring only about 1.5 centimeters in diameter on average in a skull over half a meter in length, and are hidden underneath the caniniforms, if present at all. The tooth, when present, is near the posterior and ventral edges of the maxilla, and depending on the individual may or may not extend past its medial edge,

meaning it typically did not occlude with the dentary during jaw closure. The differential growth patterns and variable presence of the tooth observed in the *Placerias* Quarry sample suggests the dismantling of the developmental pathway for the tooth in this population in the absence of selection for function. The caniniform in *Placerias* represents a particularly unusual example of functional replacement, as it not only appears to have taken over the feeding-related functions of the tooth, but also the display functions, given that it appears to have been sexually dimorphic analogously to some other tusked dicynodonts [9] and modern mammals [45].

## Speculations on behavior

Rowe [46] postulated that *Placerias* may have used their caniniforms in intraspecific combat, likening them to the sexually dimorphic horns and antlers seen in many modern ungulates. However, this interpretation was based on the version of the composite skull published in Camp and Welles [29], and did not take into account the more ventral orientation of the caniniforms suggested by Cox [14]. It also does not take into account the individual variation within large morphs, essentially only using the largest known example of the caniniform in the population to draw conclusions for the entire species. Here, our analysis suggests that the caniniform in the large morphs of *Placerias* may indeed constitute a form of sexually dimorphic weaponry, defined by Emlen [47] as a sexually selected structure that is used in physical interaction for intraspecific competition, and/or constitute sexually dimorphic ornamentation, defined by McCullough et al. [48] as being used for sexual display, without physical interaction between competing individuals. The rugose caniniforms would likely have been covered in keratin in life, and thus do not show any patterns of wear that help distinguish between these two possibilities.

Additional information can be found via comparison with living sexually dimorphic megafauna with similar features and their inferred ecological roles. Many living mammal megafauna, such as hippopotami, elephants, and walruses, have dimorphic tusks and other facial features that can be classified as weaponry [49, 50], which are similar to the caniniforms of *Placerias*. Some of these taxa (e.g., elephants) also represent species with both SSD and proportional differences in features used as weaponry, while others only differ substantially in proportional size of dimorphic weaponry and not in body size (e.g. male hippos have proportionately larger mandibles than females, and use their lower canines for intraspecific combat despite having similar body sizes) [50]. Within the large *Placerias* morph, the greater range of size in caniniforms may represent a combination of difference in ages of individuals with heterochronic indeterminate caniniform growth and individual variation based on sexual selection. This is also analogous to weaponry in modern mammal megafauna, which show the same general kind of relationship between size of the weaponry and absolute body size as is indicated by the relationship between caniniform dorsoventral length and other maxillary proportions in *Placerias*. Thus, even though none of the bones from the *Placerias* Quarry population show evidence of pre-mortem injury that could have been the result from interaction with other caniniforms [34], which is weakly suggestive of a role as ornaments rather than weapons, it seems likely that they were used as weapons irrespective of the role as ornaments.

Kinematic work on the jaw of *Placerias* has posited that it, along with other Late Triassic dicynodonts, was specialized for vertical head movement, based on the distances between occipital muscle attachment points, which correlate to proportions of neck muscles in life [51]. This inferred vertical plane of movement correlates well with multiple speculated behaviors in *Placerias*, such as rooting using the caniniforms, but also, for large morphs, aiming their ventrally projecting caniniforms anteriorly towards other individuals while holding up their

heads, in a similar manner to walruses [49]. Some of the largest *Placerias* maxillae diverge laterally, a phenomenon also observed in male walruses. While there has not been much work exploring the advantage of tusk divergence in male walruses, it may be related to the greater use of tusks as social organs in males than in females, both to enhance visual threat displays by increasing apparent tusk size [49], and potentially by increasing physical range of the tips of their tusks in combat with other males. These functions may also apply to the caniniforms in *Placerias*, both for threat displays and physical confrontations, using the cone shaped tips of the caniniforms to stab at opponents. The majority of analogous living synapsid megafaunal species that have cranial secondary sexual structures use them as weaponry. This, in combination with the similarity in robustness and shape in many of these structures to the caniniform in large morph *Placerias*, suggests that it used its caniniforms as a form of dimorphic weaponry, perhaps in addition to visual display. Assuming that sexually dimorphic tusks in other dicynodonts served a similar combat related purpose, this function may have also been taken over by the caniniform. This would make the caniniform process one of the oldest examples of sexually dimorphic weaponry in a megafaunal taxon currently known, and represent a unique form of social organ for intraspecific competition within dicynodonts.

## Supporting information

**S1 Table. Measurements of maxillae, premaxillae, quadrates, and fibulae.**
(ZIP)

**S2 Table. Parameters of measurement distributions used for likelihood calculations.**
(XLSX)

**S1 File. R script for comparing log-likelihood of unimodal and bimodal models.**
(R)

**S2 File. R script for generating PCAs and k-means clusters.**
(R)

## Acknowledgments

We thank Pat Holroyd (University of California Museum of Paleontology), Matt Smith, Bill Parker and Adam Marsh (all affiliated with Petrified Forest National Park) and Janet Gillette (Museum of Northern Arizona) for access to their museum collections, and for providing historical information on their institutions' relationships with the *Placerias* Quarry. We thank Paulo I. Prado for the R function that describes the mixture distribution. We also thank Christian Kammerer, Adam Huttenlocker, and Kenneth Angielcyzk for information on Late Triassic dicynodonts, and members of the Marshall Lab and Leah Kahn for feedback on the manuscript.

## Author Contributions

**Conceptualization:** James L. Pinto, Charles R. Marshall, Sterling J. Nesbitt, Daniel Varajão de Latorre.

**Data curation:** James L. Pinto.

**Formal analysis:** James L. Pinto, Daniel Varajão de Latorre.

**Investigation:** James L. Pinto, Sterling J. Nesbitt, Daniel Varajão de Latorre.

**Methodology:** James L. Pinto, Daniel Varajão de Latorre.

**Project administration:** James L. Pinto, Charles R. Marshall, Sterling J. Nesbitt.

**Software:** James L. Pinto, Daniel Varajão de Latorre.

**Supervision:** Charles R. Marshall, Sterling J. Nesbitt.

**Visualization:** James L. Pinto, Sterling J. Nesbitt, Daniel Varajão de Latorre.

**Writing – original draft:** James L. Pinto, Charles R. Marshall, Sterling J. Nesbitt, Daniel Varajão de Latorre.

**Writing – review & editing:** James L. Pinto, Charles R. Marshall, Sterling J. Nesbitt, Daniel Varajão de Latorre.

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
