## [Decision Letter · Decision Letter 0]

9 Oct 2023

PONE-D-23-26578Quantitative evidence for dimorphism suggests sexual selection in the maxillary caniniform process of *Placerias hesternus*PLOS ONE

Dear Dr. Pinto,

Thank you for submitting your manuscript to PLOS ONE. After careful consideration, we feel that it has merit but does not fully meet PLOS ONE’s publication criteria as it currently stands. Therefore, we invite you to submit a revised version of the manuscript that addresses the points raised during the review process.

We look forward to receiving your revised manuscript.

Kind regards,

Jun Liu

Academic Editor

PLOS ONE

Journal Requirements:

2. In your manuscript, please provide additional information regarding the specimens used in your study. Ensure that you have reported human remain specimen numbers and complete repository information, including museum name and geographic location. 

For more information on PLOS ONE's requirements for paleontology and archeology research, see https://journals.plos.org/plosone/s/submission-guidelines#loc-paleontology-and-archaeology-research.

CRM was partially supported by the Philip Sandford Boone Chair in Paleontology at the University of California, Berkeley.

Reviewers' comments:

Reviewer's Responses to Questions

**Comments to the Author**

1. Is the manuscript technically sound, and do the data support the conclusions?

Reviewer #1: Yes

2. Has the statistical analysis been performed appropriately and rigorously? 

Reviewer #1: Yes

3. Have the authors made all data underlying the findings in their manuscript fully available?

Reviewer #1: Yes

4. Is the manuscript presented in an intelligible fashion and written in standard English?

Reviewer #1: Yes

5. Review Comments to the Author

Reviewer #1: The manuscript is solid at its core, and presents a good case for sexual dimorphism in Placerias. However, the paper could use some tweaking and clarification, as suggested in my review and annotated PDF.

6. PLOS authors have the option to publish the peer review history of their article (what does this mean?). If published, this will include your full peer review and any attached files.

Reviewer #1: **Yes: **Corwin Sullivan

---

## [Author Response · Author response to Decision Letter 0]

13 Jan 2024

Dear reviewers,

Thank you for the attentive and constructive comments on our manuscript. We address your suggestions one by one in the attached Response to Reviewers document, and we think they contributed greatly to improve the clarity of our paper. We would also like to note that, while double checking values and results in the review process, we caught a minor mistake in our previous R code, which was repeated for the analysis of multiple characters. Specifically, we were using a command that didn't provide the correct sample size for the calculation of AICc. Correcting that mistake led to changes in model comparison, as can be observed in Table 1 of the manuscript document with tracked-changes. However, those changes were minor and did not affect the interpretation of the results. Figures were also slightly modified by reviewer suggestion, and exported through the PACE tool to ensure they met PLoS requirements.

---

## [Editor Report · Decision Letter 1]

16 Jan 2024

Quantitative evidence for dimorphism suggests sexual selection in the maxillary caniniform process of *Placerias hesternus*

PONE-D-23-26578R1

Dear Dr. Pinto,

We’re pleased to inform you that your manuscript has been judged scientifically suitable for publication and will be formally accepted for publication once it meets all outstanding technical requirements.

Kind regards,

Jun Liu

Academic Editor

PLOS ONE
---

## [Editor Report · Acceptance letter]

4 Apr 2024

PONE-D-23-26578R1 

PLOS ONE

Dear Dr. Pinto, 

I'm pleased to inform you that your manuscript has been deemed suitable for publication in PLOS ONE. Congratulations! Your manuscript is now being handed over to our production team.

Kind regards, 

on behalf of

Dr. Jun Liu 

Academic Editor

PLOS ONE